# *KRAS*-dependent sorting of miRNA to exosomes

**Diana J Cha[1,2†], Jeffrey L Franklin[3,4,5†], Yongchao Dou[6], Qi Liu[6], James N Higginbotham[3,4], Michelle Demory Beckler[4], Alissa M Weaver[3,7,8], Kasey Vickers[9], Nirpesh Prasad[10], Shawn Levy[10], Bing Zhang[6], Robert J Coffey[3,4,5]\*, James G Patton[1,2]\***

[1]Department of Biological Sciences, Vanderbilt University Medical Center, Nashville, United States; [2]Vanderbilt University, Nashville, United States; [3]Department of Cell and Developmental Biology, Vanderbilt University Medical Center, Nashville, United States; [4]Department of Medicine, Vanderbilt University Medical Center, Nashville, United States; [5]Affairs Medical Center, Nashville, United States; [6]Department of Biomedical Informatics, Vanderbilt University Medical Center, Nashville, United States; [7]Department of Cancer Biology, Vanderbilt University Medical Center, Nashville, United States; [8]Department of Pathology, Microbiology and Immunology, Vanderbilt University Medical Center, Nashville, United States; [9]Department of Cardiology, Vanderbilt University Medical Center, Nashville, United States; [10]HudsonAlpha Institute for Biotechnology, Huntsville, United States

**\*For correspondence:** robert. coffey@vanderbilt.edu (RJC); james.g.patton@vanderbilt. edu (JGP)

[†]These authors contributed equally to this work

**Competing interests:** The authors declare that no competing interests exist.

**Abstract** Mutant *KRAS* colorectal cancer (CRC) cells release protein-laden exosomes that can alter the tumor microenvironment. To test whether exosomal RNAs also contribute to changes in gene expression in recipient cells, and whether mutant *KRAS* might regulate the composition of secreted microRNAs (miRNAs), we compared small RNAs of cells and matched exosomes from isogenic CRC cell lines differing only in *KRAS* status. We show that exosomal profiles are distinct from cellular profiles, and mutant exosomes cluster separately from wild-type *KRAS* exosomes. *miR-10b* was selectively increased in wild-type exosomes, while *miR-100* was increased in mutant exosomes. Neutral sphingomyelinase inhibition caused accumulation of *miR-100* only in mutant cells, suggesting *KRAS*-dependent miRNA export. In Transwell co-culture experiments, mutant donor cells conferred *miR-100*-mediated target repression in wild-type-recipient cells. These findings suggest that extracellular miRNAs can function in target cells and uncover a potential new mode of action for mutant *KRAS* in CRC.

## Introduction

An emerging paradigm in the study of cell signaling is the potential role for post-transcriptional gene regulation by extracellular RNAs. microRNAs (miRNAs) are perhaps the best characterized class of small noncoding RNAs (ncRNAs) that have been detected in extracellular fluids (*Valadi et al., 2007*). Mature miRNAs are 21–23 nucleotides in length and bind to target mRNAs to inhibit their expression (*Krol et al., 2010*). Because miRNAs imperfectly pair with their mRNA targets, they can potentially regulate hundreds of transcripts within a genome (*Bartel and Chen, 2004*). However, individual miRNAs exhibit exquisite tissue-specific patterns of expression (*Wienholds et al., 2005*), control cell fate decisions (*Alvarez-Garcia and Miska, 2005*), and are often aberrantly expressed in human cancers (*Thomson et al., 2006*), affording possible disease-specific signatures with diagnostic, prognostic, and therapeutic potential (*Lu et al., 2005*; *Volinia et al., 2006*).

**eLife digest** Cells use several different methods to control which genes are expressed to produce the proteins and RNA molecules that they need to work efficiently. The first step of gene expression is to transcribe a gene to form an RNA molecule. Protein-coding mRNA molecules can then be translated to make proteins. However, many RNA transcripts do not encode proteins. One example of these non-coding RNAs is a class of small RNAs called microRNAs (miRNAs), which are predicted to target more than 60% of protein-coding genes and can control which proteins are made.

It was once thought that miRNAs exist only within the cell where they are synthesized. Recently, however, miRNAs have been found outside the cell bound to lipids and proteins, or encased in extracellular vesicles, such as exosomes. Exosomes are small bubble-like structures used by cells to export material into the space outside of cells. Exosomes containing miRNAs can circulate throughout the body, potentially transferring information between cells to alter gene expression in recipient cells.

Many colorectal cancer cells have mutations in a gene that encodes a protein called KRAS. In 2011 and 2013, researchers found that the contents of the exosomes released from these mutant *KRAS* colorectal cancer cells can influence normal cells in ways that would help a cancer to spread. Furthermore, the exosomes released from the *KRAS* mutant cells contain different proteins than non-mutant cells.

Now, Cha, Franklin et al.—including several researchers who worked on the 2011 and 2013 studies—show that exosomes released by mutant *KRAS* cells also contain miRNAs, and that these miRNAs are different from the ones exported in exosomes by cells with a normal copy of the *KRAS* gene. In particular, several miRNAs that suppress cancer growth in a healthy cell are found at lower levels in mutant KRAS cells. Instead, these miRNAs are highly represented in the exosomes that are released by the *KRAS* mutant cells.

When cells with a normal copy of the *KRAS* gene were exposed to the contents of the exosomes released from *KRAS* mutant cells, an important gene involved in cell growth was suppressed. This indicates that the miRNAs exported from cancerous cells can influence gene expression in neighboring cells. Getting rid of such cancer-suppressing miRNAs could give cancer cells a growth advantage over normal cells to promote tumor growth. Cha, Franklin et al. also suggest that it might be possible to create a non-invasive test to detect colorectal cancer by monitoring the levels of circulating miRNAs in patients. Potential treatments for the disease could also target these miRNAs.

In addition to their intracellular roles, recent experiments have identified miRNAs outside the cell in extracellular vesicles (EVs) including exosomes or larger vesicles (*Valadi et al., 2007*; *Crescitelli et al., 2013*), in high-density lipoprotein particles (*Vickers et al., 2011*), or in smaller complexes with Argonaute 2 protein (*Arroyo et al., 2011*). Exosomes are small 40–130 nm vesicles of endosomal origin that are secreted by all cells and can fuse and be internalized by recipient cells (*Valadi et al., 2007*; *Kosaka et al., 2010*; *Higginbotham et al., 2011*; *Mittelbrunn et al., 2011*; *Montecalvo et al., 2012*). It has been suggested that protein cargo transfer by exosomes between cells is associated with tumor aggressiveness and metastasis (*Skog et al., 2008*; *Higginbotham et al., 2011*; *Luga et al., 2012*; *Hoshino et al., 2013*; *Costa-Silva et al., 2015*). With the discovery that miRNAs and other RNAs can also be packaged into EVs, or exported by other extracellular mechanisms, it remains unclear the extent to which RNA cargo is sorted for export and how it is dysregulated in disease conditions, such as cancer.

Despite accumulating evidence that exosomes are biologically active, little is known regarding how oncogenic signaling affects the repertoire of miRNAs or proteins that are selected for secretion. Given the potential of cancer-derived secreted RNAs to modulate the tumor microenvironment, elucidation of the potential mechanisms for selective sorting of cargo into exosomes is critical to understanding extracellular signaling by RNA. *KRAS* mutations occur in approximately 34–45% of colon cancers (*Wong and Cunningham, 2008*). We have previously shown that exosomes from mutant *KRAS* colorectal cancer (CRC) cells can be transferred to wild-type cells to induce cell growth and migration (*Higginbotham et al., 2011*; *Demory Beckler et al., 2013*). Compared to exosomes derived from

isogenically matched wild-type cells, exosomes derived from mutant *KRAS* cells contain dramatically different protein cargo (*Demory Beckler et al., 2013*). Here, we show that *KRAS* status also prominently affects the miRNA profile in cells and their corresponding exosomes. Exosomal miRNA profiles are distinct from cellular miRNA patterns, and exosomal miRNA profiles are better predictors of *KRAS* status than cellular miRNA profiles. Furthermore, we show that cellular trafficking of miRNAs is sensitive to neutral sphingomyelinase (nSMase) inhibition in mutant, but not wild type, *KRAS* cells and that transfer of miRNAs between cells can functionally alter gene expression in recipient cells.

## Results

### Small ncRNAs are differentially distributed in exosomes

Because small RNAs are thought to be sorted at endosomal membranes and since *KRAS* signaling can also occur on late endosomes (*Lu et al., 2005*), we hypothesized that oncogenic KRAS signaling could alter RNA export into exosomes. We prepared small RNA libraries from both exosomes and whole cells using isogenically matched CRC cell lines that differ only in *KRAS* status (*Figure 1—source data 1*) (*Shirasawa et al., 1993*). Exosomes were purified using differential centrifugation and consisted of vesicles ranging in size from 40 to 130 nm (*Higginbotham et al., 2011*; *Demory Beckler et al., 2013*). These preparations exclude larger microvesicles but contain smaller lipoproteins and probably other small RNA–protein complexes (unpublished observation). Comprehensive sequencing analyses of both cellular and exosomal small RNAs from all three cell lines revealed that more than 85% of the reads from the cellular RNA libraries mapped to the genome, compared to only 50–71% from the exosomal libraries (*Figure 1A*). The non-mappable reads consisted largely of sequences that contain mismatches to genomic sequences.

The global small RNA profiles identified reads from various classes of RNA, including miRNAs, with differential enrichment of specific RNAs in both the cellular and exosomal fractions. Compared to cellular RNA samples, which displayed an enrichment of miRNA sequences (~70%), miRNA reads in exosomal samples comprised a smaller percentage of the total small reads (5–18%) compared to other ncRNA classes (e.g., tRNAs, rRNAs, snRNAs) (*Figure 1B,C*, *Supplementary file 1*). Most of these reads appear to be the fragments of larger RNAs, both cytoplasmic and nuclear. It is not clear how these RNAs are associated with and/or deposited into exosomes.

The size distribution of cellular small RNA matched that expected from miRNA-derived reads (21–23 nucleotides). However, the small RNA read distribution from exosomes was much broader with many reads smaller than 22 nucleotides in length (*Figure 1—figure supplement 1*). Given that these reads map to RNAs other than known miRNAs, these data suggest that a large proportion of small exosomal RNA reads is derived from processing of other RNAs, in addition to post-transcriptionally modified miRNA reads that are apparently subject to editing, trimming, and/or tailing (*Koppers-Lalic et al., 2014*). Consistent with this, when read identity was restricted to miRNAs by mapping back to known miRNA hairpin sequences, the length distribution of mappable reads was nearly identical between cells and exosomes (*Figure 1D*).

### miRNAs are differentially enriched in exosomes dependent on *KRAS* status

Focusing on mappable reads, we sought to ascertain whether miRNAs might be differentially represented when comparing cells to their secreted exosomes. For this, we quantified the relative abundance of individual miRNAs and made pairwise comparisons between normalized miRNA reads. Spearman correlation analyses demonstrated high correlation between replicates of individual cell lines (r = 0.95–0.96) and between cellular data sets differing only in *KRAS* status (r = 0.92–0.96) (*Figure 2—figure supplements 1–3*). In contrast, the miRNA profiles from exosomes compared to their parent cells were less correlated (DKO-1 r = 0.67–0.81, DKs-8 r = 0.64–0.71, DLD-1 r = 0.64–0.69) (*Figure 2—figure supplements 1, 2, 4*).

We next utilized principal component (PC) analysis to determine whether the overall miRNA profiles could distinguish between cells and exosomes and/or between wild-type and mutant *KRAS* status. The miRNA profiles from the three cell lines all clustered close to one another indicating that overall miRNA expression profiles are fairly similar among the different cell types (*Figure 2*). In marked contrast, PC analysis revealed that exosomal miRNA profiles clearly segregate according to *KRAS* status (*Figure 2*). Relatively, minor differences between cellular miRNA expression profiles become

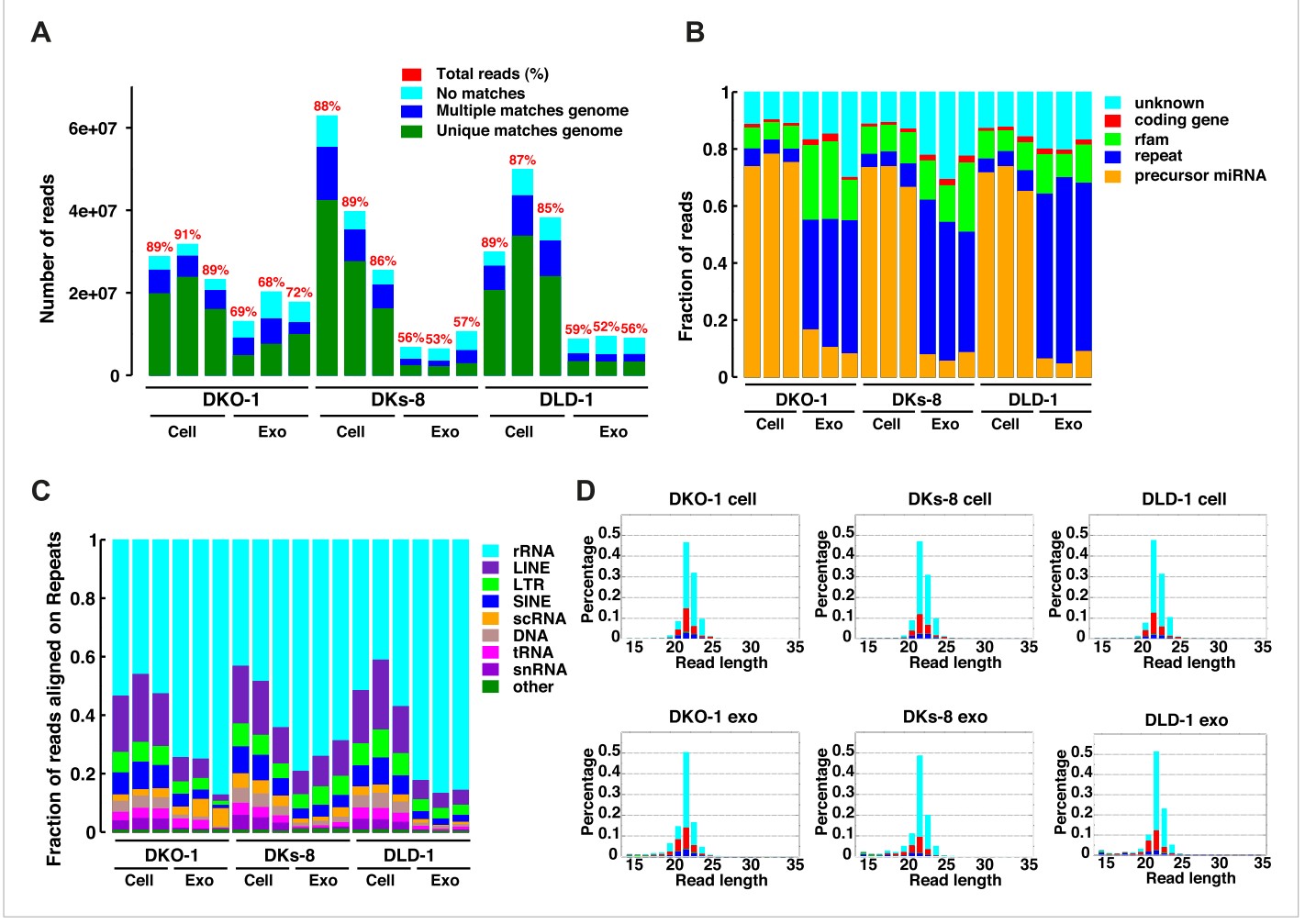

**Figure 1**. Small RNA sequencing analysis of cellular and exosomal RNAs from CRC cell lines. Shown are (**A**) total read numbers (y-axis) and the total percentage of mappable reads (red), percentage of unique mappable reads (green), reads that map to multiple genomic locations (dark blue), and those that could not be mapped (cyan). (**B**) The majority of mappable small RNA reads were derived from noncoding RNAs in cells and exosomes. In cells, the majority of small RNA reads mapped to microRNAs (miRNAs) (miRbase 19), whereas in exosomes, the majority of small RNA reads mapped to repetitive elements. (**C**) The origin of repetitive reads from exosomal small RNA sequencing is shown. Repeat reads were annotated based on RepeatMasker and Rfam classified into tRNAs, rRNAs, snRNAs, and others. (**D**) The length distribution of reads mapping to miRNA hairpins was determined for small RNA reads from the three CRC cell lines and their purified exosomes. Colors represent the nucleotides identified for the 5′ base, T (cyan), A (red), G (green), and C (blue). *Figure 1—figure supplement 1*.

The following source data and figure supplement are available for figure 1:

**Source data 1**. Colorectal cancer (CRC) cell lines.

**Figure supplement 1**. Length distribution of small RNA reads to genome.

much more prominent when comparing exosomal miRNA patterns (*Figure 2—figure supplement 3*). This indicates that the presence of a mutant *KRAS* allele alters sorting of specific miRNAs to exosomes, a finding that has potentially important implications for biomarker development.

To gain more insight into the relative abundance of miRNAs in cells vs matched exosomes, we examined the most abundant miRNA species in the various sequencing libraries (determined by mean reads of individual miRNAs). For many miRNAs, exosomal abundance correlated with cellular abundance (*Supplementary file 2*). However, calculation of fold changes among the three isogenic *KRAS* cell lines, and exosomes released from these cells, showed that distinct subsets of miRNAs are

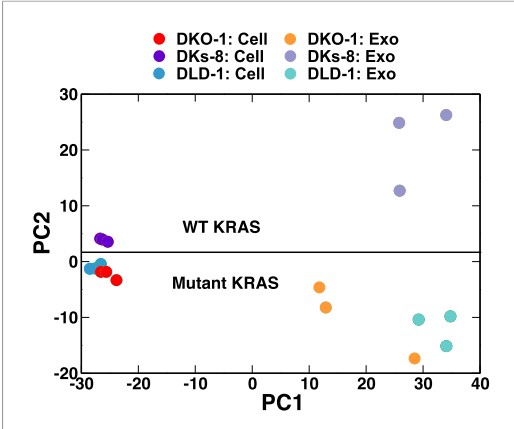

**Figure 2**. Small RNA composition segregates with KRAS status. Principal component analysis was performed comparing small RNA sequencing data sets from CRC cells and exosomes. The small RNA composition from cells differed significantly from exosomes. Nevertheless, clustering showed that mutant *KRAS* status could be inferred from small RNA composition. Also see *Figure 2—figure supplements 1–4*.

The following figure supplements are available for figure 2:

**Figure supplement 1**. Spearman correlations between miRNA expression profiles in cells and exosomes.

**Figure supplement 2**. Spearman correlations between miRNA expression profiles in cells and exosomes.

**Figure supplement 3**. Spearman correlations between miRNA expression profiles in cells (top) and exosomes (bottoms) in reads per million (RPM).

**Figure supplement 4**. Spearman correlations comparing miRNA expression profiles in exosomes to parent cell in RPM.

enriched in either exosomes or cells (*Tables 1, 2*). For all three cell lines, 25 miRNAs were consistently upregulated in cells and 29 miRNAs were consistently upregulated in exosomes (*Figure 3A,B*). Additionally, the diversity of miRNAs was substantially greater in mutant *KRAS* DKO-1 exosomes (94 unique miRNAs) compared to parental DLD-1 or wild-type *KRAS* DKs-8 exosomes (*Figure 3B*). A select subset of cell and exosomally targeted miRNAs were validated separately by quantitative reverse-transcription PCR (qRT/PCR) (*Figure 3C*). Collectively, these data indicate that the miRNA profiles observed in exosomes are distinct from their parental cells with specific miRNAs preferentially overrepresented or underrepresented in exosomes. We observed a mutant *KRAS*-specific pattern of secreted miRNAs, consistent with the hypothesis that dysregulation of miRNA metabolism is associated with tumorigenesis, a previously unrecognized feature of mutant *KRAS*.

## KRAS-dependent sorting of miRNAs

### miR-100

Down regulation of *miR-100-5p* was observed in mutant *KRAS* DKO-1 and parental DLD-1 cells compared to wild-type *KRAS* DKs-8 cells (*Table 1*). This is consistent with reports that have shown decreased *miR-100* expression in metastatic cancers (*Petrelli et al., 2012*; *Gebeshuber and Martinez, 2013*). *miR-100* has also been shown to negatively regulate migration, invasion, and the epithelial–mesenchymal transition (EMT) (*Chen et al., 2014*; *Wang et al., 2014*; *Zhou et al., 2015*). Interestingly, *miR-100* was enriched in exosomes derived from mutant *KRAS* cells (>eightfold and >threefold enriched in DKO-1 and DLD-1 exosomes, respectively; *Supplementary file 2*), suggesting that decrease of *miR-100* in cells is due to secretion in exosomes. This is in line with findings that circulating levels of *miR-100* are upregulated in the plasma of mutant *KRAS*-expressing mouse pancreatic cancer models and in patients with pancreatic cancer (*LaConti et al., 2011*). More broadly, the observation that *miR-100-5p* specifically accumulates in exosomes suggests that there may be sequence-specific requirements for the sorting of certain miRNAs into exosomes.

### miR-10b

Our RNA sequencing data identified *miR-10b* as preferentially secreted in exosomes isolated from cells harboring a wild-type *KRAS* allele (>threefold-change and >twofold-change enrichment in DKs-8 and DLD-1 exosomes, respectively) but retained in mutant *KRAS* DKO-1 cells (~threefold-change cell enrichment). *miR-10b* is referred to as an oncomiR because it is frequently upregulated during progression of various cancers, including CRC (*Ma, 2010*).

### miR-320

*miR-320* is aberrantly expressed in several types of cancer, including colon cancer. It is expressed in the proliferative compartment of normal colonic crypts (*Schepeler et al., 2008*; *Hsieh et al., 2013*).

**Table 1**. Differential expression of miRNAs in colorectal cancer cells*

DKO-1

| | | | |
|---|---|---|---|
| hsa-miR-548u | hsa-miR-16-1-3p | hsa-miR-33a-3p | hsa-miR-33a-5p |
| hsa-miR-31-5p | hsa-miR-181b-3p | hsa-miR-450a-5p | hsa-miR-424-5p |
| hsa-miR-9-5p | hsa-miR-219-5p | hsa-miR-190a | hsa-miR-573 |
| hsa-miR-30d-3p | hsa-miR-204-5p | hsa-miR-1226-3p | hsa-miR-499a-5p |
| hsa-miR-450b-5p | hsa-miR-499b-3p | hsa-miR-3662 | hsa-miR-20a-3p |
| hsa-miR-27b-5p | hsa-miR-5701 | hsa-miR-4677-3p | hsa-let-7i-5p |
| hsa-miR-331-3p | hsa-miR-31-3p | hsa-miR-651 | hsa-miR-1306-5p |
| hsa-miR-147b | hsa-miR-3611 | hsa-miR-1305 | hsa-miR-148a-3p |
| hsa-miR-27b-3p | hsa-miR-1306-3p | hsa-miR-374b-3p | hsa-miR-1260b |
| hsa-miR-3940-3p | hsa-miR-200c-5p | hsa-miR-548ar-3p | |

DKs-8

| | | | |
|---|---|---|---|
| hsa-miR-132-5p | hsa-miR-484 | hsa-miR-374a-5p | hsa-miR-1180 |
| hsa-miR-1307-3p | hsa-miR-200a-5p | hsa-miR-548o-3p | hsa-miR-149-5p |
| hsa-miR-3615 | hsa-miR-100-5p | hsa-miR-197-3p | hsa-miR-378a-5p |
| hsa-let-7a-3p | | | |

DLD-1

| | | | |
|---|---|---|---|
| hsa-miR-141-3p | hsa-miR-26b-5p | hsa-miR-24-3p | hsa-miR-3074-5p |
| hsa-miR-15a-5p | hsa-miR-27a-3p | hsa-miR-3613-5p | hsa-miR-30b-5p |
| hsa-miR-29a-3p | hsa-miR-301a-5p | hsa-let-7i-3p | hsa-miR-185-5p |
| hsa-let-7g-5p | hsa-miR-23b-3p | hsa-miR-22-3p | |

DKO-1 and DKs-8

| | | | |
|---|---|---|---|
| hsa-miR-141-5p | hsa-miR-582-5p | | |

DKO-1 and DLD-1

| | | | |
|---|---|---|---|
| hsa-miR-556-3p | hsa-miR-374a-3p | hsa-miR-106b-5p | hsa-miR-17-3p |
| hsa-miR-24-1-5p | hsa-miR-340-3p | | |

DLD-1 and DKs-8

| | | | |
|---|---|---|---|
| hsa-miR-24-2-5p | hsa-miR-106a-5p | hsa-miR-30e-5p | hsa-miR-107 |
| hsa-miR-429 | hsa-miR-98-5p | hsa-miR-425-5p | hsa-miR-140-5p |
| hsa-miR-93-5p | hsa-miR-210 | hsa-miR-126-3p | hsa-miR-194-5p |
| hsa-miR-29b-3p | hsa-miR-15b-5p | hsa-miR-362-5p | hsa-miR-27a-5p |
| hsa-miR-454-3p | hsa-miR-452-5p | hsa-miR-196b-5p | |

DKO-1, DLD-1 and DKs-8

| | | | |
|---|---|---|---|
| hsa-miR-32-5p | hsa-miR-582-3p | hsa-miR-542-3p | hsa-miR-96-5p |
| hsa-miR-101-3p | hsa-miR-18a-5p | hsa-miR-3529-3p | hsa-miR-7-5p |
| hsa-miR-19a-3p | hsa-miR-142-3p | hsa-miR-20a-5p | hsa-miR-32-3p |
| hsa-miR-130b-5p | hsa-miR-1278 | hsa-miR-7-1-3p | hsa-miR-590-3p |
| hsa-miR-4473 | hsa-miR-17-5p | hsa-miR-103a-3p | hsa-miR-103b |
| hsa-miR-19b-3p | hsa-miR-340-5p | hsa-miR-200a-3p | hsa-miR-34a-5p |
| hsa-miR-372 | | | |

*miRNAs differentially enriched in cells when comparing mean reads in exosomes vs cell.

miRNAs expression patterns were compared between DKs-8, DKO-1, and DLD-1 cells. miRNAs were identified that were enriched in just one of the three cell types or that overlapped between combinations of cells. For miRNAs, 25 were identified that are expressed in all three cell types, 13 were enriched in DKs-8 cells, 15 in DLD-1 cells, and 39 were unique to DKO-1 cells.

**Table 2**. Differential distribution of miRNAs in exosomes*

DKO-1

| | | | |
|---|---|---|---|
| hsa-miR-139-5p | hsa-miR-3178 | hsa-miR-151b | hsa-miR-125b-1-3p |
| hsa-miR-193b-3p | hsa-miR-935 | hsa-miR-130b-3p | hsa-miR-628-3p |
| hsa-miR-139-3p | hsa-let-7d-3p | hsa-miR-589-3p | hsa-miR-4532 |
| hsa-miR-451a | hsa-miR-6087 | hsa-miR-151a-5p | hsa-miR-940 |
| hsa-miR-222-3p | hsa-miR-766-5p | hsa-miR-505-5p | hsa-miR-3187-3p |
| hsa-miR-125a-3p | hsa-miR-3679-5p | hsa-miR-4436b-3p | hsa-miR-4787-3p |
| hsa-miR-2277-3p | hsa-miR-361-5p | hsa-miR-1293 | hsa-miR-3183 |
| hsa-miR-3162-5p | hsa-miR-642a-3p | hsa-miR-642b-5p | hsa-miR-197-5p |
| hsa-miR-324-3p | hsa-miR-145-3p | hsa-miR-3182 | hsa-miR-3127-3p |
| hsa-miR-3127-5p | hsa-miR-4728-3p | hsa-miR-3184-5p | hsa-miR-125b-5p |
| hsa-miR-186-5p | hsa-miR-1 | hsa-miR-100-5p | hsa-miR-423-3p |
| hsa-miR-766-3p | hsa-miR-4753-5p | hsa-miR-145-5p | hsa-miR-4724-5p |
| hsa-miR-373-3p | hsa-miR-223-5p | hsa-miR-1307-5p | hsa-miR-1914-3p |
| hsa-miR-3121-3p | hsa-miR-3613-3p | hsa-miR-205-5p | hsa-miR-98-3p |
| hsa-miR-23a-3p | hsa-miR-3124-5p | hsa-miR-3656 | hsa-miR-3918 |
| hsa-miR-4449 | hsa-miR-378c | hsa-miR-3138 | hsa-miR-1910 |
| hsa-miR-3174 | hsa-miR-4466 | hsa-miR-3679-3p | hsa-miR-3200-5p |
| hsa-miR-6511b-5p | hsa-miR-1247-5p | hsa-miR-22-3p | hsa-miR-877-5p |
| hsa-miR-4687-3p | hsa-miR-1292-5p | hsa-miR-181c-5p | hsa-miR-6131 |
| hsa-miR-6513-5p | hsa-miR-3661 | hsa-miR-132-3p | hsa-miR-214-3p |
| hsa-miR-574-3p | hsa-miR-3190-3p | hsa-miR-326 | hsa-miR-3191-5p |
| hsa-miR-3198 | hsa-miR-3928 | hsa-miR-629-3p | hsa-miR-4489 |
| hsa-miR-4700-5p | hsa-miR-5006-5p | hsa-miR-5088 | hsa-miR-2110 |
| hsa-miR-3911 | hsa-miR-3146 | | |

DKs-8

| | | | |
|---|---|---|---|
| hsa-miR-1224-5p | hsa-let-7b-5p | hsa-miR-155-5p | hsa-let-7c |
| hsa-let-7a-5p | hsa-miR-146b-5p | hsa-miR-4647 | hsa-miR-4494 |
| hsa-miR-711 | hsa-miR-1263 | | |

DLD-1

| | | | |
|---|---|---|---|
| hsa-miR-1226-5p | hsa-miR-4745-5p | hsa-miR-4435 | hsa-miR-939-5p |
| hsa-miR-409-3p | hsa-miR-1304-3p | | |

DKO-1 and DKs-8

| | | | |
|---|---|---|---|
| hsa-miR-146a-5p | hsa-miR-4508 | hsa-miR-224-5p | hsa-miR-4429 |
| hsa-miR-222-5p | hsa-miR-629-5p | hsa-miR-4492 | hsa-miR-3653 |
| hsa-miR-320a | hsa-miR-1290 | hsa-miR-1262 | hsa-miR-5010-5p |
| hsa-miR-204-3p | hsa-miR-4461 | hsa-miR-5187-5p | |

DKO-1 and DLD-1

| | | | |
|---|---|---|---|
| hsa-miR-483-5p | hsa-miR-4658 | hsa-miR-4758-5p | hsa-miR-492 |
| hsa-miR-5001-5p | hsa-miR-371a-5p | hsa-miR-1323 | hsa-miR-371b-3p |
| hsa-miR-501-3p | hsa-miR-4446-3p | hsa-miR-6511a-5p | hsa-miR-30a-3p |
| hsa-miR-4727-3p | | | |

DLD-1 and DKs-8

| | | | |
|---|---|---|---|
| hsa-miR-28-3p | hsa-miR-3934-5p | | |

DKO-1, DLD-1 and DKs-8

*Table 2. Continued on next page*

*Table 2. Continued*

| | | | |
|---|---|---|---|
| hsa-miR-658 | hsa-miR-320d | hsa-miR-4792 | hsa-miR-1246 |
| hsa-miR-320e | hsa-miR-4516 | hsa-miR-320b | hsa-miR-4488 |
| hsa-miR-1291 | hsa-miR-320c | hsa-miR-4634 | hsa-miR-3605-5p |
| hsa-miR-4741 | hsa-miR-3591-3p | hsa-miR-122-5p | hsa-miR-486-3p |
| hsa-miR-184 | hsa-miR-223-3p | hsa-miR-3651 | hsa-miR-486-5p |
| hsa-miR-3180 | hsa-miR-3180-3p | hsa-miR-3168 | hsa-miR-4497 |
| hsa-miR-423-5p | hsa-miR-3184-3p | hsa-miR-150-5p | hsa-miR-664a-5p |
| hsa-miR-182-5p | | | |

*miRNAs differentially enriched in exosomes when comparing mean reads in exosomes vs cell.
miRNAs expression patterns were compared between exosomes purified from DKs-8, DKO-1, and DLD-1 cells. miRNAs were identified that were enriched in exosomes from just one of the three cell lines or that overlapped between combinations of cell lines. 29 miRNAs were common between exosomes from all three cell lines. 94 were enriched in exosomes from DKO-1 cells, 10 in exosomes from DKs-8 cells, and only 6 in exosomes from DLD-1 cells.

miR-320 members (miR-320b, -c, d, and -e) were abundant in both mutant KRAS (DKO-1) and wild-type KRAS (DKs-8) exosomes, but underrepresented in the matched cells, indicating that some miRNAs are transcribed and predominantly exported into exosomes, independent of KRAS status (Table 2, Supplementary file 1A). Of these family members, miR-320a and miR-320b were the most abundant species represented in exosomes by our RNA sequencing analyses (miR-320a in DKO-1 exosomes, and miR-320b in DKs-8 and DKO-1 exosomes). Interestingly, however, we observed the largest enrichment for miR-320d (fold changes >241 in DKs-8 and >229 in DKO-1) in exosomes relative to cells, despite being ~fourfold less abundant than miR-320b levels. Because the 3′-terminus may be important in regulating miRNA stability and turnover, coupled to the fact that the sequences of miR-320a-d members differ only at their 3′-termini, enrichment of certain miRNAs in exosomes could be due to higher turnover/decay rates in cells.

## Exosomal secretion and strand selection

Because we observed differential export of specific miRNAs, we investigated whether there might be miRNA sequence-specific sorting signals. Previous reports have shown differential accumulation of 5p or 3p strands in exosomes compared to parental cells (Ji et al., 2014). Thus, we analyzed our data sets to test whether exosomes might be preferentially enriched for one strand over the other. We were able to identify individual miRNAs where the two strands differentially sorted between cells and exosomes. For example, the -5p strands of miR-423 were overrepresented in DKO-1 exosomes but in exosomes from DKs-8 cells, both strands were overrepresented compared to cells (data not shown). This indicates that KRAS status may differentially affect selection of passenger or guide strands for sorting into exosomes for select individual miRNAs.

Individual miRNAs often exist as populations of variants (isomiRs) that differ in length and/or nucleotide composition generated by template- or non-template-directed variation (Burroughs et al., 2010; Newman et al., 2011; Neilsen et al., 2012). When we analyzed our sequencing data sets, we did not detect differential accumulation of isomers with variable 5′ termini (data not shown). For cellular miRNAs, most reads were full length with a slight enrichment in 3′ non-templated addition of A-tailed miRNAs, regardless of KRAS status (Figure 4; Figure 4—figure supplement 1). For exosomes, we observed a slight enrichment for C residues added to the 3′ ends of miRNAs from wild-type KRAS cells (Figure 4—figure supplement 1). We did not observe this in mutant KRAS exosomes, where instead, we noticed an increase in 3′ trimming of miRNAs (Figure 4, Figure 4—figure supplement 2). Overall, it remains to be determined whether such modifications constitute a global exosomal sorting signal in these cells.

Consistent with published data, we have shown that miRNA expression patterns vary between parental cells and their cognate exosomes (Tables 1, 2, Figure 3A,B) (Valadi et al., 2007; Mittelbrunn et al., 2011; Ekstrom et al., 2012; Montecalvo et al., 2012; Squadrito et al., 2014). Differential export suggests that specific signals must exist to sort distinct miRNAs (Batagov et al.,

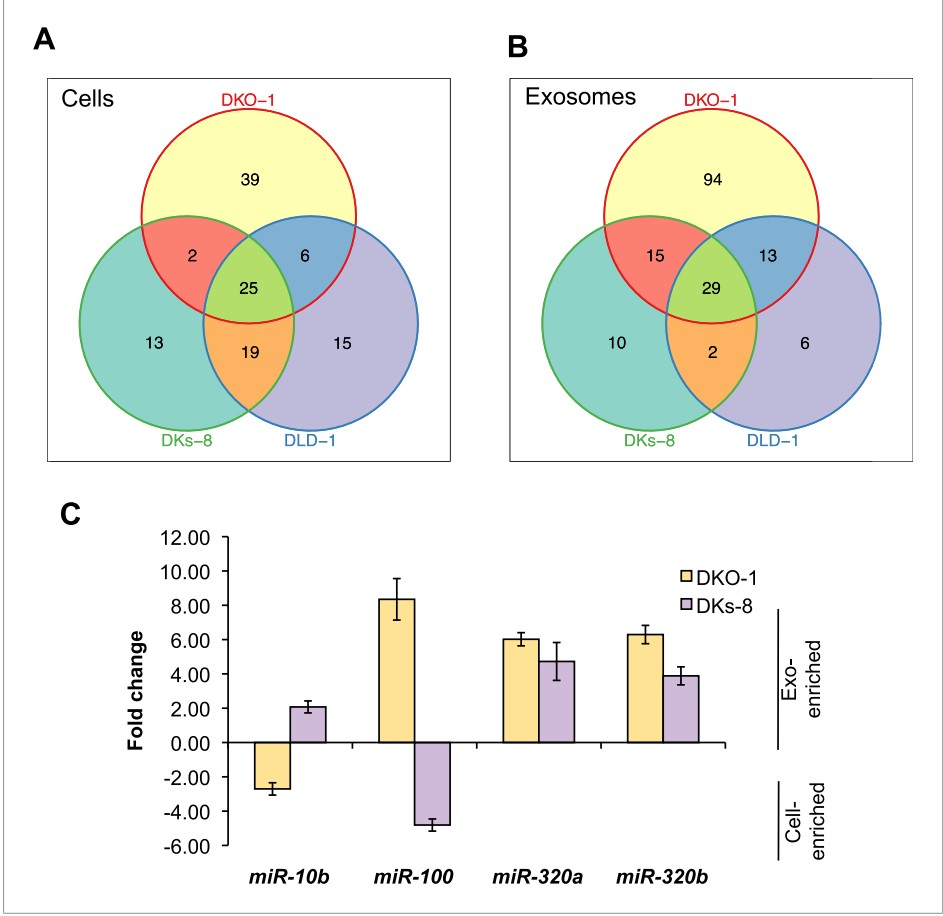

**Figure 3**. *KRAS*-dependent regulation of miRNAs in exosomes and cells. Differentially distributed miRNAs in (**A**) cells and (**B**) exosomes from the three CRC cell lines differing in *KRAS* status. (**C**) qRT-PCR validation of selected miRs from DKs-8 and DKO-1 cellular and exosomal RNA samples normalized to U6 snRNA. Fold changes were calculated using the $\Delta\Delta C(t)$ method comparing exosomes to cells. Negative fold changes indicate greater enrichment in cells, and positive fold changes indicate greater enrichment in exosomes. Also see *Supplementary file 2*.

*2011*; *Villarroya-Beltri et al., 2013*). We therefore conducted MEME analysis to attempt to identify sequence motifs that might serve as targeting signals. When we examined all miRNA reads detected in exosomes, we did not find any global enrichment for specific sequences or motifs, including those reported to be bound by hnRNP A2B1 (GGAG or U/CC) (*Bolukbasi et al., 2012*; *Villarroya-Beltri et al., 2013*) (*Figure 4—figure supplement 3*). However, when we analyzed *miR-320* because it is preferentially exported to exosomes independent of *KRAS* status, we were able to identify the GGAG sequence contained within the 3′ end of the mature sequence. Additionally, upon restricting our analysis to reads from the most differentially expressed miRNAs when comparing exosomes to cells, we found a slight enrichment for C residues, possibly alternating C residues in exosomal miRNAs (*Figure 4—figure supplement 3*).

## Sphingomyelinase-dependent sorting of miRNAs to exosomes

Although little is understood regarding the molecular mechanisms for packaging exosomal miRNAs, recent evidence suggests that the secretion of miRNAs in exosomes is dependent on ceramide via its production by neutral sphingomyelinase 2 (nSMase 2) (*Kosaka et al., 2010*; *Mittelbrunn et al., 2011*). Inhibition of de novo ceramide synthesis by treatment with a nSMase inhibitor impaired exosomal miRNA release, apparently due to decreased formation of miRNA-containing exosomes

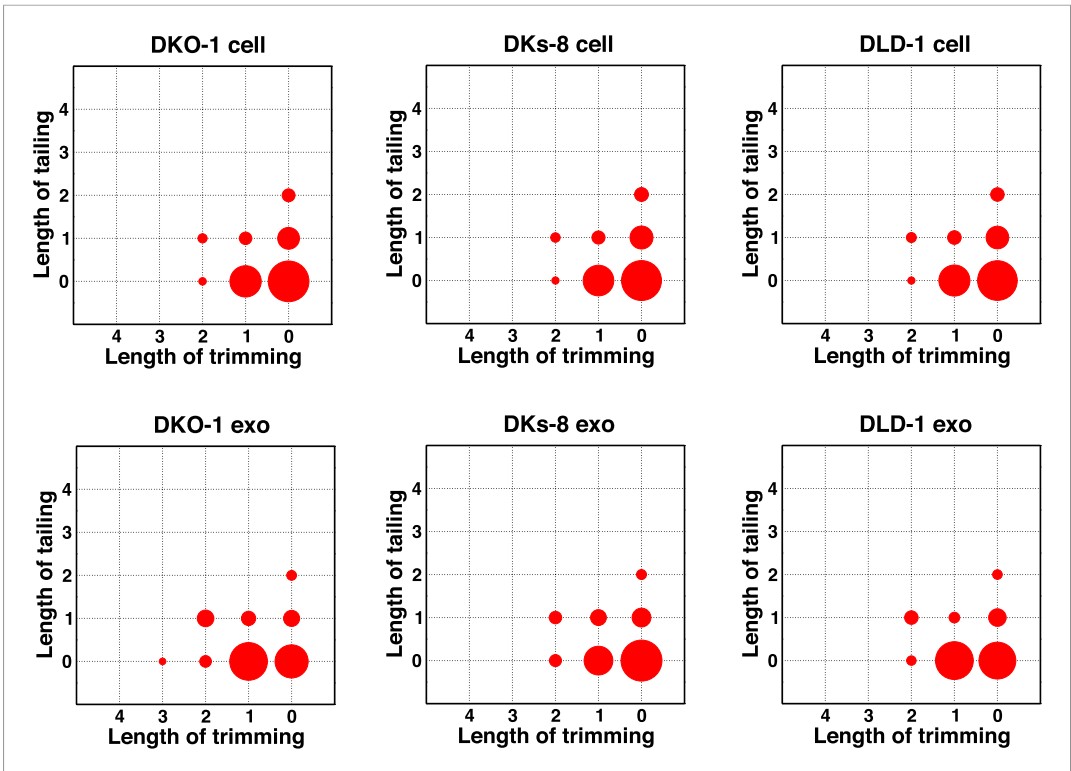

**Figure 4**. Comparison of miRNA 3′ trimming and tailing between cells and exosomes. Data from the heat maps shown in *Figure 4—figure supplement 2* were pooled to illustrate overall changes in either 3′ nucleotide additions (tailing) or 3′ resection (trimming) compared to full length miRNA sequences (intact). Overall, the patterns between cells and between exosomes are very similar. A comparison of cells to exosomes shows that exosomes display a slight increase in trimmed miRNAs. Also see *Figure 4—figure supplement 2*.

The following figure supplements are available for figure 4:

**Figure supplement 1**. Non-templated addition (NTA) of nucleotides to 3′ ends of miRNAs.

**Figure supplement 2**. Comparison of miRNA 3′ trimming and tailing between cells and exosomes.

**Figure supplement 3**. MEME analysis of miRNA sequence in exosomes.

(*Kosaka et al., 2010*; *Mittelbrunn et al., 2011*). To test the role of nSMase in miRNA secretion in our system, we treated CRC cells with the nSMase inhibitor, GW4869. We determined the effect of this inhibitor on *miR-10b* since it is preferentially found in wild-type *KRAS* DKs-8 exosomes, *miR-100* since it is preferentially found in mutant *KRAS* DKO-1 and DLD-1 exosomes, and *miR-320* which sorts into exosomes regardless of *KRAS* status. For *miR-10b*, we did not observe significant changes in its cellular levels after treatment with GW4869 in either wild-type *KRAS* DKs-8 or mutant *KRAS* DKO-1 cells (*Figure 5C*). In contrast, inhibition of nSMase caused a ∼threefold increase in intracellular levels of *miR-100* in mutant *KRAS* DKO-1 cells but remained unchanged in wild-type DKs-8 *KRAS* cells (*Figure 5A,B,C*). Similarly, *miR-320* levels were found to increase (∼2.5 fold) only in GW4869-treated mutant *KRAS* DKO-1 cells (*Figure 5C*). These data are most consistent with the hypothesis that impaired ceramide synthesis alters cellular accumulation of miRNAs dependent on mutant *KRAS* and suggest that multiple biogenic routes exist for miRNA secretion.

## Extracellular transfer of *miR-100*

Several reports have found that extracellular miRNAs can be taken up by recipient cells to mediate heterotypic cell–cell interactions and facilitate target repression in neighboring cells

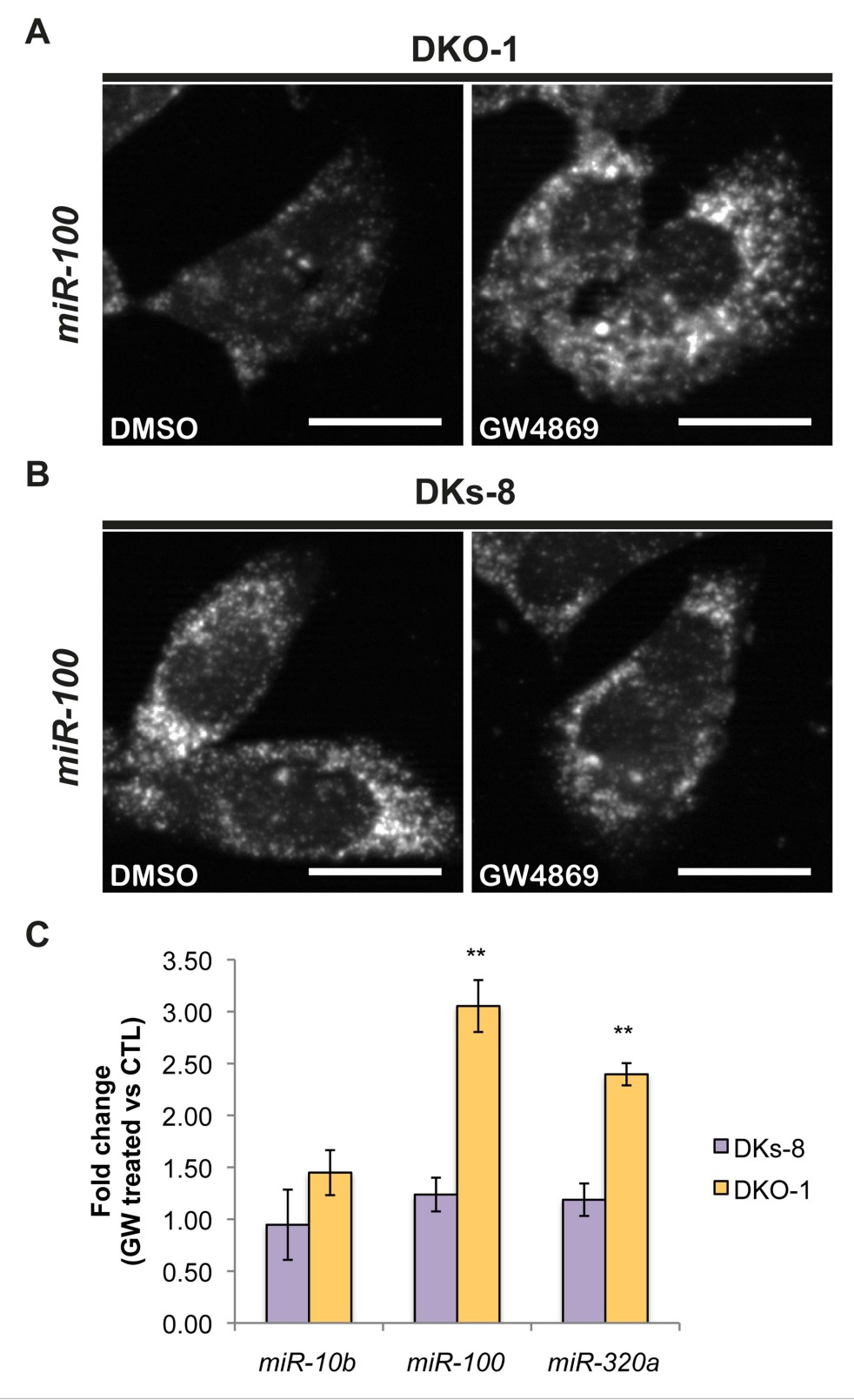

**Figure 5**. Ceramide-dependent miRNA export into exosomes. DKO-1 or DKs-8 cells were treated with an inhibitor of neutral sphingomyelinase 2 (nSMase 2), GW4869. After treatment, in situ hybridization experiments were performed with probes against *miR-100* (**A**, **B**). (**C**) qRT-PCR for *miR-10b, miR-100,* and *miR-320a* was performed on
*Figure 5. continued on next page*

*Figure 5. Continued*

cells treated with GW4869 or DMSO, and fold change in expression was determined in treated vs untreated cells. In wild-type *KRAS* cells (DKs-8), inhibition of nSMase 2 had little or no effect on the cellular levels of these miRNAs. In contrast, mutant *KRAS* cells (DKO-1) showed an increase in cellular miRNA levels after inhibition of nSMAse 2. Data were derived from three biological replicates and performed in technical triplicates for qRT-PCR. Significance was determined by two-tailed, paired t-tests where * are p values ≤ 0.05 and ** ≤0.01.

(*Mittelbrunn et al., 2011*; *Boelens et al., 2014*; *Squadrito et al., 2014*). To determine whether secreted miRNAs function in recipient cells, we designed luciferase (Luc) constructs containing either 3 perfect *miR-100* recognition elements (MREs) in the 3′ untranslated region (UTR) (Luc-100-PT) or scrambled 3′UTR sequences that do not match any known miRNAs (Luc-CTL). These constructs were expressed in wild-type *KRAS* DKs-8 cells (recipient cells) in the presence or absence of donor cells. Baseline repression of Luc in the absence of donor cells was first analyzed to determine the levels of repression from endogenous *miR-100* in DKs-8 cells. Compared to the scrambled control (Luc-CTL), strong Luc repression in the absence of donor cells was observed with perfect MREs (*miR-100*-PT) (*Figure 6A*). This supports our finding that *miR-100* is expressed and retained in DKs-8 cells.

To determine whether secretion of *miR-100* by mutant *KRAS* DKO-1 donor cells could further augment *miR-100* function in recipient wild-type cells, Transwell co-culture experiments were performed with DKs-8 recipient cells expressing the Luc reporters in the presence of DKO-1 donor cells (*Figure 6*). Significantly increased repression of Luc was observed when the reporter construct containing three perfect *miR-100* sites was used (*miR-100*-PT) (*Figure 6A*). Because exosomes released from DKO-1 cells contain abundant levels of *miR-100,* increased Luc repression is consistent with transfer of additional copies of *miR-100.* Two control experiments were performed to test the hypothesis that additional copies of *miR-100* are transferred between donor and recipient cells. First, we treated donor cells with antagomirs that block production of *miR-100.* Luc repression was almost completely reversed upon pre-treatment of DKO-1 donor cells with a *miR-100* hairpin antagomir inhibitor (AI-100) (*Figure 6D*). Second, we performed qRT/PCR to calculate the increase in *miR-100* levels in recipient cells. Cells grown in the presence or absence of donor cells showed an approximate 34% increase in the levels of *miR-100* (*Figure 6E* and *Figure 6—figure supplement 2*).

To further probe the repressive activity of *miR-100*, we performed co-culture experiments in which the recipient Dks-8 cells express Luc fused to the 3′UTR of mTOR, an endogenous *miR-100* target (*Nagaraja et al., 2010*; *Grundmann et al., 2011*; *Ge et al., 2014*). As observed with *miR-100*-PT repression, Luc-mTOR was significantly repressed in the presence of mutant *KRAS* DKO-1 but not in the presence of wild-type *KRAS* DKs-8 donor cells (*Figure 6B*). This suggests that *miR-100*-repressive activity is specific to the presence of mutant *KRAS* DKO-1 donor cells. To confirm these results, we mutated the MREs within the mTOR 3′UTR and assayed for *miR-100* activity (*Figure 6—figure supplement 1*). Mutation of individual sites did not show significantly different Luc repression (data not shown). However, upon mutation of two MREs (MS2), we observed a partial rescue of Luc expression (*Figure 6C*). This was further augmented upon mutation of all three sites (MS3), with a complete rescue of *miR-100*-mediated repressive activity (*Figure 6C*).

As a final test of miRNA transfer in the Transwell co-culture experiments, we created vectors expressing Luc fused to a 3′UTR containing perfect sites for *miR-222* because *miR-222* is not detectable in DKs-8-recipient cells, unlike *miR-100.* In this case, silencing of Luc should be due to transfer of *miR-222* and not due to unforeseen changes in endogenous miRNA activity. We observed a greater than twofold repression of the *miR-222* Luc reporter in recipient cells (*Figure 6—figure supplement 3*). These results support the hypothesis that miRNAs secreted by mutant *KRAS* cells can be transferred to recipient cells.

## Discussion

In this study, we comprehensively examined the composition of small ncRNAs from exosomes and cells of isogenic CRC cell lines that differ only in *KRAS* status. By employing small RNA transcriptome analyses, we found that oncogenic *KRAS* selectively alters the miRNA profile in exosomes, and that ceramide depletion selectively promotes miRNA accumulation in mutant *KRAS* CRC cells. Distinct miRNA profiles between cells and their exosomes may be functionally coupled to mitogenic signaling.

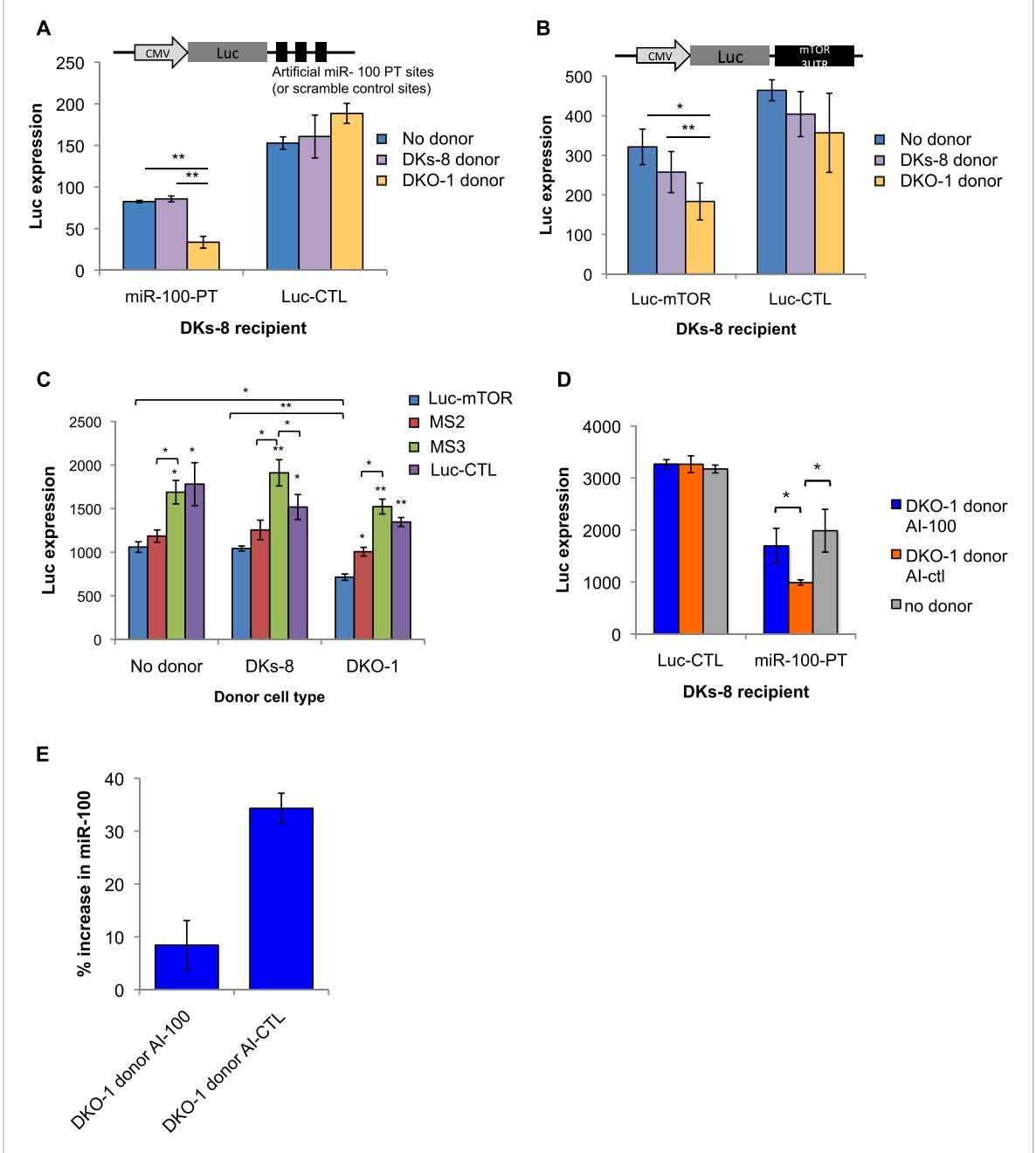

**Figure 6.** Transfer of extracellular miRNAs by mutant DKO-1 cells promotes target repression in wild-type DKs-8 cells. Transwell co-culture of DKs-8 recipient cells with or without DKs-8 or DKO-1 donor cells. Luciferase (Luc) expression was measured in DKs-8 recipient cells transiently expressing (**A**) Luc fused to three perfectly complementary synthetic *miR-100* target sites (*miR-100*-PT) or (**B**) Luc fused to the 3′UTR of mTOR, which harbors 3 endogenous target sites for *miR-100*. (**C**) Luc expression increased upon mutation of two (MS2) sites with full expression upon mutation of all three sites (MS3). Luc-CTL contains three random scrambled target sites that do not match any known miRNA sequence. (**D**) Luc expression was restored in recipient cells expressing *miR-100*-PT upon pretreatment of donor DKO-1 cells with 100 nM *miR-100* antagomirs (AI-100) compared to pre-treatment of donor DKO-1 cells with 100 nm control antagomirs (AI-CTL) targeting *cel-miR-67*. (**E**) Taqman qRT-PCR for *miR-100*. Compared to DKs-8 recipient cells grown without donor cells, *mir-100* levels increased by approximately 34% in the presence of mutant DKO-1 donor cells pre-treated with AI-CTL compared to an 8% increase in AI-100 pre-treated donor cells. Y axis is % increase in $miR\text{-}100 = (CP_{AI\text{-}CTL}$ or $CP_{AI\text{-}100} - CP_{no\ donor}/CP_{no\ donor})*100$, where CP = absolute copy number. All Luc values were normalized to co-transfected vectors expressing β-galactosidase; n = 3 independent experiments in **A**–**C** and n = 4 in **D**, **E**. All Luc assays were performed in technical triplicate. Significance was determined by two-tailed, paired t-tests where * are p values ≤ 0.05 and ** ≤0.01. Also see *Figure 6—figure supplements 1–3*.

*Figure 6. continued on next page*

*Figure 6. Continued*

The following figure supplements are available for figure 6:

**Figure supplement 1**. *miR-100* binding sites in the mTOR 3′UTR.

**Figure supplement 2**. Presence of mutant DKO-1 donor cells augments *miR-100* levels in DKs-8 recipient cells.

**Figure supplement 3**. Transfer of extracellular miRNAs by mutant DKO-1 cells promotes target repression in wild-type DKs-8 cells.

*KRAS* status-specific patterns of secreted miRNAs support the idea of using exosomes as potential biomarkers in CRC. Our finding that *miR-10b* is preferentially enriched in wild-type *KRAS*-derived exosomes, while *miR-100* is enriched in mutant *KRAS*-derived exosomes raises interesting questions regarding how they are selected for secretion. *miR-10b* and *miR-100* are both part of the *miR-10/100* family and differ by only one base in the seed region, allowing regulation of distinct sets of target mRNAs (*Tehler et al., 2011*). Whether the accumulation or export of these miRNAs is a result or a consequence of oncogenic signaling remains unknown. Preventing the export or retention of certain miRNAs, such as *miR-100 and miR-10b,* may serve a therapeutic role in reversing the tumorigenic effects seen with aberrant miRNA expression.

*KRAS*-dependent differential miRNA expression more prominently affected miRNA expression patterns observed in exosomes than in the parent cells. This could reflect a mechanism by cells to selectively export miRNAs so as to maintain specific growth or gene expression states. This is consistent with a recent report that found that the cellular levels of *miR-218-5p* could be maintained, despite changes in the abundance of its target, likely through a 'miRNA relocation effect' where unbound miRNAs that are in excess have the potential to be sorted to exosomes (*Squadrito et al., 2014*). Another mechanism may be through sequence-specific motifs that direct miRNA trafficking by interaction with specific chaperone proteins (*Bolukbasi et al., 2012*; *Villarroya-Beltri et al., 2013*). Although we did not find any globally significant motif overrepresented in exosomal miRNAs, we cannot rule out that individual miRNAs might undergo sequence-specific export. *miR-320* family members all contain the GGAG motif that has been proposed to serve as an exosomal targeting signal (*Villarroya-Beltri et al., 2013*). We found that members of the *miR-320* family are preferentially enriched in exosomes independent of *KRAS* status; however, the GGAG sequence was not found in other miRNAs that are targeted to exosomes. It has been reported that the biogenesis of *miR-320* family members occurs by a non-canonical pathway that requires neither Drosha (*Chong et al., 2010*) nor XPO5 (*Xie et al., 2013*). Instead, the 5′ ends contain a 7-methyl guanosine cap that facilitates nuclear–cytoplasmic transport through XPO1 (*Xie et al., 2013*). XPO1 is present in DKO-1, DKs-8, and DLD-1 exosomes as detected by mass spectrometry (*Demory Beckler et al., 2013*). It will be interesting to investigate whether alternate processing pathways and associated biogenic machinery contribute to the heterogeneity of EV cargo and affect miRNA secretion.

It was recently demonstrated that miRNAs in B-cell exosomes display enriched levels of non-template-directed 3′-uridylated miRNAs, while 3′-adenylated miRNA species are preferentially cell enriched (*Koppers-Lalic et al., 2014*). In certain contexts, the addition of non-templated uridine residues to cognate miRNAs accelerates miRNA turnover (*Baccarini et al., 2011*; *Wei et al., 2012*). Thus, it is possible that the stability/half-life of a miRNA affects whether it is retained or secreted. While the exact functional significance of 3′-end modifications of miRNAs detected in both cells and exosomes remains to be determined, it could be that differential export of 'tagged' miRNAs could allow cells to export specific miRNAs. However, the lack of any apparent motif upon global analysis of miRNAs enriched in exosomes, coupled to the finding that even untagged miRNAs are differentially exported, suggests multiple strategies for loading of miRNAs into EVs, and that not all EVs and exosomes contain identical cargo. This further implies that different cell types secrete a heterogeneous population of vesicles. Although the biological relevance of these findings remains to be determined, the specific sorting of miRNAs into exosomes may enable cancer cells to discard tumor-suppressive miRNAs so as to increase their oncogenic potential or perhaps modulate gene expression in neighboring and distant cells to promote tumorigenesis. In support of this hypothesis, *miR-100*, which we found to be enriched in mutant *KRAS* exosomes, was found to down-regulate LGR5 in CRC

cells and thereby inhibit migration and invasion of such cells (*Zhou et al., 2015*). In this context, removal of *miR-100* from the cell would be a tumor-promoting event.

In other contexts, *miR-100* can have contradictory activities, both inducing EMT by down-regulating E-cadherin through targeting SMARCA5 and inhibiting tumorigenicity by targeting HOXA1 (*Chen et al., 2014*). Thus, although *miR-100* can function as a tumor suppressor under normal conditions, augmenting its levels, for example, by EV uptake, could potentially promote EMT. In this regard, the role of *miR-100* in tumorigenesis would be twofold, where its secretion in exosomes could function to maintain low-intracellular levels within mutant cells, while inducing EMT in wild-type-recipient cells. Along these lines, *miR-100* is part of the *miR-125b/let-7a-2/miR-100* cluster that is transcribed and expressed coordinately (*Emmrich et al., 2014*). Interestingly, in malignant colonic tissues from individuals with CRC, *miR-100* levels were significantly decreased while *let-7a* levels were strongly upregulated (*Tarasov et al., 2014*). Based on our finding that there is differential accumulation of individual miRNAs within this cluster between mutant *KRAS* cells and exosomes, it will be interesting to determine whether cancer cells down-regulate specific miRNAs by active secretion, while simultaneously maintaining the levels of other miRNAs transcribed within the same cluster.

miRNAs are secreted from malignant breast epithelial cells after packaging into vesicles larger than conventional exosomes that are enriched in CD44, whose expression is linked to breast cancer metastasis (*Palma et al., 2012*). Normal cells tend to release miRNAs in more homogenous types of exosomes, suggesting that malignant transformation may alter the formation of secreted vesicles that could alter miRNA export and lead to differences in exosome content and morphology (*Palma et al., 2012*; *Melo et al., 2014*). In support of this, it was recently shown that in exosomes from breast cancer cells, CD43 mediates the accumulation of Dicer (*Melo et al., 2014*). These exosomes also contain other RNA-induced silencing complex (RISC) proteins and pre-miRNAs, indicating that miRNA processing can occur in exosomes (*Melo et al., 2014*). These components were absent in exosomes derived from normal cells. It remains to be determined whether components of the RISC-loading complex assemble within endosomes before their secretion as exosomes or by the fusion of exosomes containing heterogeneous cargo after they are secreted. The observation that cells can selectively release miRNAs and also release a heterogeneous population of vesicles raises the possibility that differential release of miRNAs is associated with different classes of exosomes and microvesicles.

Recently, quantitative analysis of secreted miRNAs suggested that the levels of extracellular miRNAs are limited and raise the question as to how such levels can alter gene expression in recipient cells (*Chevillet et al., 2014*). The results of our Transwell co-culture experiments are most consistent with extracellular transfer of specific miRNAs to alter expression of reporter constructs. Nevertheless, the level of exosomal transfer that is needed to alter recipient cell gene expression in vivo remains an open question. Our finding that mutant *KRAS* protein can be functionally transferred in exosomes indicates that the full effect of exosomes on recipient cells can be due to a combination of both RNA delivery and protein-based signaling (*Higginbotham et al., 2011*). This could include activation of Toll-like receptors with possible downstream effects following nuclear factor kappa-light-chain-enhancer of activated B cells or mitogen-activated protein kinase cascades (*Fabbri et al., 2012*; *Chen et al., 2013*). The complexity of *miR-100* function in the tumor microenvironment underscores this argument by its potential for inhibiting mTOR expression which is required for proliferation of *Apc*-deficient tumors in mouse models (*Faller et al., 2015*). In tumors where some cells have incurred activating mutations in *KRAS*, while others have not, *miR-100* could accumulate in wild-type *KRAS* tumor cells through exosomal transfer, inhibiting mTOR and cell growth. Conversely, *miR-100* could be secreted from mutant *KRAS* cells giving them a growth advantage. In this way, exosomal transfer of miRNAs might act to select for cells carrying specific tumor driver mutations. Our studies have direct implications for CRC and, together with other studies, indicate that delivery of exosomes to recipient cells can induce cell migration, inflammation, immune responses, angiogenesis, invasion, pre-metastatic niche formation, and metastasis (*Kahlert and Kalluri, 2013*; *Boelens et al., 2014*; *Melo et al., 2014*; *Costa-Silva et al., 2015*).

## Materials and methods

### Exosome isolation

Exosomes were isolated from conditioned medium of DKO-1, Dks-8, and DLD-1 cells as previously described, with slight modification (*Higginbotham et al., 2011*). Briefly, cells were cultured in Dulbecco's Modified Eagle's Medium (DMEM) supplemented with 10% bovine growth serum until

80% confluent. The cells were then washed three times with Phosphate buffered saline (PBS) and cultured for 24 hr in serum-free medium. The medium was collected and replaced with ionomycin-containing medium for 1 hr, after which ionomycin-containing medium was collected and pooled with the previously collected serum-free medium. Pooled media was centrifuged for 10 min at 300×$g$ to remove cellular debris, and the resulting supernatant was then filtered through a 0.22-mm polyethersulfone filter (Nalgene, Rochester, NY, USA) to reduce microparticle contamination. The filtrate was concentrated ~300-fold with a 100,000 molecular weight cut-off centrifugal concentrator (Millipore, Darmstadt, Germany). The concentrate was then subjected to high-speed centrifugation at 150,000×$g$ for 2 hr. The resulting exosome-enriched pellet was resuspended in PBS containing 25 mM hydroxyethyl-piperazineethanesulfonic acid (HEPES) (pH 7.2) and washed by centrifuging again at 150,000×$g$ for 3 hr. The wash steps were repeated a minimum of three times until no trace of phenol red was detected. The resulting pellet was resuspended in PBS containing 25 mM HEPES (pH 7.2), and protein concentrations were determined with a MicroBCA kit (Pierce/Thermo, Rockford, IL, USA). The number of exosomes per µg of protein was determined by means of nanoparticle tracking analysis (NanoSight, Wiltshire, UK). Analysis was performed on three independent preparations of exosomes.

## RNA purification

Total RNA from exosomes and cells was isolated using TRIzol (Life Technologies/Thermo, Grand Island, NY). In the case of exosomal RNA isolation, TRIzol was incubated with 100 µl or less of concentrated exosomes for an extended 15 min incubation prior to chloroform extraction. RNA pellets were resuspended in 60 µl of RNase-free water and were then re-purified using the miRNeasy kit (Qiagen Inc., Valencia, CA, USA). Final RNAs were eluted with two rounds of 30 µl water extraction.

## miRNA library preparation and sequencing

Total RNA from each sample was used for small RNA library preparation using NEBNext Small RNA Library Prep Set from Illumina (New England BioLabs Inc., Ipswich, MA, USA). Briefly, 3′ adapters were ligated to total input RNA followed by hybridization of multiplex single read (SR) reverse transcription (RT) primers and ligation of multiplex 5′ SR adapters. RT was performed using ProtoScript II RT for 1 hr at 50°C. Immediately after RT reactions, PCR amplification was performed for 15 cycles using LongAmp Taq 2× master mix. Illumina-indexed primers were added to uniquely barcode each sample. Post-PCR material was purified using QIAquick PCR purification kits (Qiagen Inc.). Post-PCR yield and concentration of the prepared libraries were assessed using Qubit 2.0 Fluorometer (Invitrogen, Carlsbad, California, CA, USA) and DNA 1000 chip on Agilent 2100 Bioanalyzer (Applied Biosystems, Carlsbad, CA, USA), respectively. Size selection of small RNA with a target size range of approximately 146–148 bp was performed using 3% dye free agarose gel cassettes on a Pippin Prep instrument (Sage Science Inc., Beverly, MA, USA). Post-size selection yield and concentration of libraries were assessed using Qubit 2.0 Fluorometer and DNA high-sensitivity chip on an Agilent 2100 Bioanalyzer, respectively. Accurate quantification for sequencing applications was performed using qPCR-based KAPA Biosystems Library Quantification kits (Kapa Biosystems, Inc., Woburn, MA, USA). Each library was diluted to a final concentration of 1.25 nM and pooled in equimolar ratios prior to clustering. Cluster generation was carried out on a cBot v8.0 using Illumina's Truseq Single Read Cluster Kit v3.0. Single-end sequencing was performed to generate at least 15 million reads per sample on an Illumina HiSeq2000 using a 50-cycleTruSeq SBSHSv3 reagent kit. Clustered flow cells were sequenced for 56 cycles, consisting of a 50-cycle read, followed by a 6-cycle index read. Image analysis and base calling were performed using the standard Illumina pipeline consisting of Real Time Analysis version v1.17 and demultiplexed using bcl2fastq converter with default settings.

## Mapping of RNA reads

Read sequence quality checks were performed by FastQC (Babraham Bioinformatics [http://www.bioinformatics.babraham.ac.uk/projects/fastqc/]). Adapters from the 3′ ends of reads were trimmed using Cutadpt with a maximum allowed error rate of 0.1 (*Martin, 2011*). Reads shorter than 15 nucleotides in length were excluded from further analysis. Reads were mapped to the human genome hg19 using Bowtie version 1.1.1 (*Langmead and Salzberg, 2012*). Mapped reads were annotated using ncPRO-seq (*Chen et al., 2012*) based on miRbase (*Griffiths-Jones et al., 2008*), Rfam (*Gardner et al., 2011*; *Burge et al., 2013*), and RepeatMasker (http://www.repeatmasker.org/), and expression

levels were quantified based on read counts. Mature miRNA annotation was extended 2 bp in both upstream and downstream regions to accommodate inaccurate processing of precursor miRNAs. Reads with multiple mapping locations were weighted by the number of mapping locations.

## PC analysis

DESeq Version 1.16.0 was used to perform PC analyses (*Anders and Huber, 2010*).

## Enrichment analysis

Differential expression was analyzed using DESeq Version 1.16.0 (*Anders and Huber, 2010*). Negative binomial distribution was used to compare miRNA abundance between cells vs exosomes and wild-type vs mutant *KRAS* status. The trimmed mean of M values method was used for normalization (*Robinson and Oshlack, 2010*). Differential expression was determined based on log2 fold change (log2 fold change) and false discovery rate (FDR) with |log2 fold change| $\geq$ 1 and FDR $\leq$ 0.001.

## Trimming and tailing

Trimming and tailing analysis was based on miRBase annotation (*Griffiths-Jones et al., 2006*, *2008*; *Griffiths-Jones, 2010*). Only high-confidence miRNAs (544) and corresponding hairpin sequences were used. Bowtie version 1.1.1 with 0 mismatch was used for mapping. miRNA reads were first mapped to hairpin sequences with unmapped reads, then mapped to the human genome hg19. Remaining reads were trimmed 1 bp from the 3′ end and remapped to hairpin sequences. The remapping process was repeated 10 times. Finally, all mapped reads were collected for further analysis.

## qRT/PCR

Taqman small RNA assays (Life Technologies) (individual assay numbers are listed below) were performed for indicated miRNAs on cellular and exosomal RNA samples. Briefly, 10 ng of total RNA was used per individual RT reactions; 0.67 µl of the resultant cDNA was used in 10 µl qPCR reactions. qPCR reactions were conducted in 96-well plates on a Bio-Rad CFX96 instrument. All C(t) values were $\leq$30. Triplicate C(t) values were averaged and normalized to U6 snRNA. Fold changes were calculated using the $\Delta\Delta$C(t) method, where $\Delta$ = C(t)$_{miRNA}$ − C(t)$_{U6\ snRNA}$, and $\Delta\Delta$C(t) = $\Delta$C(t)$_{exo}$ − $\Delta$C(t)$_{cell}$, and FC = $2^{-\Delta\Delta C(t)}$. Analysis was performed on three independent cell and exosomal RNA samples. Taqman probe #: U6 snRNA: 001973; *hsa-let-7a-5p*: 000377; *hsa-miR-100-5p*: 000437; *hsa-miR-320b*: 002844; hsa-miR-320a: 002277.

## Generation of miRNA standard curves

RNase-free, HPLC-purified 5′-phosphorylated miRNA oligoribonucleotides were synthesized (Integrated DNA Technologies) for human *miR-100-5p* (5′-phospho-AACCCGUAGAUCCGAACUUGUG-OH-3′) and *cel-miR-39-3p* (5′-phospho-UCACCGGGUGUAAAUCAGCUUG-OH-3′). Stock solutions of 10 µM synthetic oligonucleotide in RNase-free and DNase-free water were prepared according to the concentrations and sample purity quoted by the manufacturer (based on spectrophotometry analysis). Nine twofold dilution series beginning with 50 pM synthetic oligonucleotide were used in 10 µl RT reactions (Taqman small RNA assays), and qPCR was performed. Each dilution was performed in triplicate from three independent experiments. Linear regression was used to determine mean C(t) values plotted against log(miRNA copies/µl).

## miRNA in situ hybridizations and ceramide dependence

Cells were plated in 6-well plates containing coverslips at a density of ~2.5 × $10^5$ cells and cultured in DMEM supplemented with 10% bovine growth serum for 24 hr. The cells were then washed three times with PBS and cultured for 24 hr in serum-free medium containing either 5 µM GW4869 (Cayman Chemicals # 13127, Ann Arbor, MI, USA) or DMSO. Medium was removed and cells were washed three times with PBS and fixed with 4% Paraformaldehyde (PFA) for ~15 min at room temperature. After, cells were washed three times in DEPC-treated PBS and permeabilized in 70% ethanol for ~4 hr at 4°C, and rehydrated in DEPC-treated PBS for 5 min. Pre-hybridization was performed in hybridization buffer (25% formamide, 0.05 M EDTA, 4× saline-sodium citrate (SSC), 10% dextran sulfate, 1X Denhardt's solution 1 mg/ml *Escherichia coli* tRNA) in a humidified chamber at 60°C for 60 min. Hybridization buffer was removed and replaced with 10 nM of probe (probe numbers are listed below) diluted in hybridization buffer and incubated at either 55°C (*miR-100* and *miR-10b*) or 57°C for

scrambled and U6 probes for 2 hr. Coverslips were then washed in series with pre-heated SSC at 37°C as follows: 4× SSC briefly, 2× SSC for 30 min, 1× SSC for 30 min, and 0.1× SSC for 20 min. miRNA detection was conducted using Tyramide Signal Amplification (Perkin Elmer, # NEL741001KT, Waltham, MA, USA). Briefly, coverslips were blocked in blocking buffer (0.1 M TRIS-HCl, pH 7.5, 0.15 M NaCl, 0.5% Blocking Reagent [Roche, #11096176001, Basel, Switzerland]) at 4°C overnight. Blocking buffer was replaced with anti-DIG-POD (Roche, # 11207733910) diluted 1:100 in blocking buffer and incubated for 60 min. Coverslips were washed three times, 5 min per wash, in wash buffer (0.1 M Tris-HCl, pH 7.5, 0.15 M NaCl, 0.5% Saponin) followed by incubation with 1× Fluorescein diluted in 1× amplification reagent for 5 min. Fluorescent coverslips were then washed two times, 5 min per wash, in wash buffer. To preserve fluorescent signals, coverslips were fixed with 2% PFA containing 2% Bovine serum albumin in 1× PBS for 15 min. After fixation, coverslips were washed 2 times, 5 min per wash, in wash buffer, followed by a final wash in 1× PBS for 5 min. Coverslips were then mounted in Prolong Gold (Life Technologies) and visualized on a Zeiss LSM510 at 63× objective. 3′-DIG labeled probes for in situ hybridizations-U6 snRNA: 99002-05; Scramble: 99004-05; *miR-10b-5p*: 38486-05; *miR-100-5p*: 18009-05 (Exiqon, Woburn, MA, USA).

## Co-culture and Luc reporter assays

Recipient cells were plated in six-well plates at a density of ∼2.5 × 10$^5$ cells and cultured in DMEM supplemented with 10% bovine growth serum for 24 hr. Media was replaced and cells were co-transfected (Promega, E2311, Madison, WI, USA) with 1.5 µg of Luc-reporter plasmid and 1.5 µg β-gal plasmid DNA/well. Donor cells were plated in 0.4-µm polyester membrane Transwell filters (Corning, 3450, Corning, NY, USA) at ∼2.5 × 10$^5$ cells/well for 24 hr. Media from donor Transwells and recipient 6-well plates were removed and replaced with DMEM without FBS. Co-culture of donor and recipient cells was conducted for either 24 or 48 hr before recipient cells were harvested. Lysates were prepared in 1× Reporter lysis buffer (Promega, E2510), and Luc assays were performed according to the manufacturer's protocol (Promega, E2510). β-gal expression was simultaneously determined from the lysates according to the manufacturer's protocol (Promega, E2000). Differences in transfection efficiency were accounted for by normalizing Luc expression to β-Gal expression (Luc/β-Gal). All assays were performed on three biological replicates, each with three technical replicates.

## Antagomir treatment

Donor cells were plated in 0.4-µm polyester membrane Transwell filters (Corning, 3450, Corning, NY, USA) at ∼1.4 × 10$^4$ cells/well for 24 hr. Medium was replaced and donor cells were transfected with either *miR-100* hairpin antagomirs (# IH-300517-05, GE Life Sciences) or negative control hairpin antagomirs corresponding to *cel-miR-67* (# IN-001005-01, GE Life Sciences) to produce a final concentration of 100 nM of antagomir for 24 hr. Medium from donor Transwells and recipient 6-well plates was removed and replaced with DMEM without FBS. Co-culture of donor and recipient cells was conducted for 24 hr before recipient cells were harvested for RNA isolation.

## Plasmid construction

For the pLuc-mTOR construct, the 3′UTR of *mTOR* was PCR amplified (primer sequences in *Supplementary file 3*) from genomic DNA isolated from DKs-8 cells. The amplicon was cloned into pMiR-Report (Life Technologies) via SpeI/HinDIII restriction sites. Mutation of *miR-100* binding sites in mTOR 3′UTR (MS) was performed on pLuc-mTOR using forward or reverse primers targeting either all three MRE's, or MRE 2 and 3 with QuikChange Lightning Multi-Site Directed Mutagenesis (Agilent, Santa Clara, CA, USA) according to manufacturer's protocol. To create the reporter construct containing three *miR-100* perfect sites (miR-100-PT), oligonucleotides (*Supplementary file 3*) were annealed to produce a synthetic fragment containing the perfect sites with CTAGT and AGCTT overhangs. The fragment was cloned into pMiR-report via SpeI/HinDIII restriction sites. All plasmids were sequence verified (GeneWiz, South Plainfield, NJ, USA).

## Acknowledgements

This work was supported by grants from the National Institutes of Health, U19CA179514, RO1 CA163563 and a GI Special Program of Research Excellence (SPORE) P50 95103 to RJC, and a pilot in P30 DK058404 to JLF. Vanderbilt Digestive Disease Research Center (P30 DK058404) and associated Cores.

## Additional information

### Funding

| Funder | Grant reference | Author |
|---|---|---|
| National Institutes of Health (NIH) | U19CA179514 | Robert J Coffey, James G Patton |
| National Cancer Institute (NCI) | P50 95103 | Robert J Coffey |
| National Institutes of Health (NIH) | RO1 CA163563 | Robert J Coffey |
| National Institutes of Health (NIH) | P30 DK058404 | Jeffrey L Franklin |

The funders had no role in study design, data collection and interpretation, or the decision to submit the work for publication.

### Author contributions

DJC, JLF, Conception and design, Acquisition of data, Analysis and interpretation of data, Drafting or revising the article; YD, QL, JNH, MDB, NP, SL, Acquisition of data, Analysis and interpretation of data; AMW, KV, RJC, JGP, Conception and design, Drafting or revising the article; BZ, Conception and design, Analysis and interpretation of data, Drafting or revising the article

## Additional files

### Supplementary files

• Supplementary file 1. Read counts for (**A**) individual miRNA species and (**B**) repeat families.

• Supplementary file 2. Abundant miRNAs. Normalized read counts (see 'Materials and methods') were used to determine the miRNAs with the highest number of reads (top 50) from cell and exosomal data sets. The top 50 most abundant miRNAs were compared between exosomes and cells for each cell line. Most abundant miRNAs in exosomes (blue), cells (orange), or both.

• Supplementary file 3. Related to experimental procedures. Primers used for plasmid construction.

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
