## [Decision Letter]

Thank you for sending your work entitled “*KRAS*-Dependent Sorting of miRNA to Exosomes” for consideration at *eLife*. Your article has been favorably evaluated by Sean Morrison (Senior editor) and two reviewers, one of whom is a member of our Board of Reviewing Editors.

The Reviewing editor and the other reviewer discussed their comments before we reached this decision, and the Reviewing editor has assembled the following comments to help you prepare a revised submission.

The function of exosomal miRNAs is a highly controversial area of research, with some studies, such as this manuscript, claiming that miRNAs are not only exported from cells in exosomes, but also taken up by other cells as biologically functional signaling molecules, and others arguing they have no biological function. This manuscript, likely for the first time in the field establishes a well-defined experimental system: three isogenic cells lines that differ solely in *KRAS* status.

The manuscript makes some progress toward testing the idea that miRNAs can be transferred from cell-to-cell, but given the controversies in the field, a higher standard of proof is required to merit publication in *eLife*. We recommend the authors be given the opportunity to revise their manuscript, providing the requested new experiments and analyses. Most importantly, I urge the authors to spend less time selling their story and more effort rigorously testing-not proving-their hypotheses.

1) The major paper deficiency is lack of a clear biological model or mechanism explaining the data. While this is also true for most published exosome papers, one expects an *eLife* paper to propose some explanation for why specific miRNAs are transferred from cell-to-cell according to the exporting cell's *KRAS* status.

2) Correlation analysis plays a central role in testing the authors' hypothesis. Given the low correlation of the independent biological replicates (deep sequencing replicates typically correlate with R > 0.90 in one of our labs), the authors should apply an appropriate statistical test to determine that R-values of 0.92-0.96 between cells are unlikely to differ from R-values of 0.67-0.89 comparing exosomes to the exporting cells? If the bottom quartile of miRNAs by abundance (i.e., the ones least well measured by convention, rather than digital, sequencing methods) are excluded, do the Pearson correlation values change? Can all the biological replicates be used to make the comparisons, not simply pairwise combinations of individual data sets?

3) Is the degree of reporter repression small because the abundance of exosome-delivered miRNAs is low? The miRNA literature overwhelmingly supports the view that low abundance miRNAs have no biological effects, because the cellular concentration of miRNA-binding sites is much, much greater than the concentration of miRNA. That is, the stoichiometric mechanism of miRNA-mediated repression in mammals requires that miRNAs be highly abundant. When DKs-8 cells obtain a miRNA, such as *miR-222*, from exosomes, does that new miRNA rank in the top 25% or 50% of miRNAs by abundance? If not, it is difficult to imagine how it could be functional, given the aggregate intracellular concentration of seed-matched target sites. The authors need to report an estimate of how many molecules of a given miRNA sequence are present per exosome and how many are delivered to an individual recipient cell.

4) Why were three perfect sites used? Were controls performed validating the reporter using anti-miRs and miRNA mimics?

5) In the ceramide experiments, the authors interpret the change in exosomal and cellular abundance for *miR-100* and *miR-320* as evidence that a subset of miRNA sorting is altered by ceramide while a separate, ceramide-independent pathway delivers other miRNAs to exosomes. The data are interesting, but don't seem to contribute to our understanding of the mechanism of putative sorting of miRNAs into exosomes. Perhaps *miR-10b* is simply less abundant than *miR-100* or *miR-320*, making it harder to reliably detect changes in its abundance?

6A) High-Throughput Sequencing Data. How were the data normalized? How was the normalization procedure validated? Best practice is to select the normalization method that produces the greatest congruence among otherwise identical biologically independent replicates.

6B) Extending miRNA sequences {plus minus} 2 nt “to accommodate inaccurate processing of precursor miRNAs” would be a great idea if miRBase were always right; but miRBase is often wrong. It would be better to use the sequence of the most abundant isoform of the miRNA as the “accurately” processed form and to pool reads for all isoforms with the same 5′ end (i.e., the same seed sequence).

6C) Whenever read data is presented, species data should be presented in parallel. For example, the data in Figure 1 would have a very different meaning if most of the “repeat” sequences were from just a few species, rather than a diverse set of RNAs.

---

## [Author Response]

*1) The major paper deficiency is lack of a clear biological model or mechanism explaining the data. While this is also true for most published exosome papers, one expects an* eLife *paper to propose some explanation for why specific miRNAs are transferred from cell-to-cell according to the exporting cell's* KRAS *status*.

It remains a key question as to how specific miRNAs are selected for export. We conducted a series of experiments as detailed in the Results to determine whether any previously proposed mechanistic models can explain why we detect enrichment for some miRNAs within exosomes. We rigorously evaluated our data sets and described in the Results and the Discussion that we could not find support for:

A) “Zip-code” sequences (Bolukbasi et al. Mol Therapy-Nucleic Acids, 1 e10, Batagov et al. BMC Genomics 12: S18). These papers proposed that specific nucleotide patterns within secreted RNAs targeted them for export. We tested for the presence of those signals, as well as common motifs using MEME analysis, in our data sets and could not find evidence for a universal zip code sequence. However, when we restricted our analysis to the most differentially represented miRNAs in exosomes compared to cells, we detected a possible enrichment for C residues or alternating C residues. This is shown in Figure 4—figure supplement 3. We have added sentences to address these findings but we were careful to ensure that readers understand that the overall conclusion is that we were unable to identify a short sequence that could serve as a zip code targeting element.

B) 3’ and 5’ modifications (Koppers-Lalic et al. Cell Reports 8:1-10, Katoh et al. Genes Dev 23: 433, Burns et al. Nature 473: 105, Fernandez-Valverde et al. RNA 16: 1881, Wei et al. RNA 18: 915, Thornton et al. NAR 42: 11777). Numerous reports have proposed that 5’ and 3’ modifications can alter miRNA metabolism. Most commonly, addition of U residues to 3’ ends is thought to promote turnover whereas the addition of A residues promotes stabilization. G and C additions are generally rare. We tested whether exosomal export might be linked to 3’ or 5’ modifications. We did not observe significant 5’ modifications from reads derived from miRNAs in either cells or exosomes. For 3’ modifications, we found that cellular miRNAs tended to have increased numbers of modified 3’ ends with added A residues. We did not observe an enrichment of 3’-NTA of A residues in reads derived from exosomal miRNAs. Further analysis showed that there was enrichment of reads containing extra C residues at the 3’ ends in exosomes from wild-type KRAS cells. These data are now included in Figure 4—figure supplement 1. It remains unclear whether these changes are key to providing mechanistic insight into miRNA export. We have added sentences to carefully address this point.

C) Sumoylated hnRNPA2 B1 (Villarroya-Bletri et al. Nature Comm 4:2980). This paper proposed that miRNAs destined for exosomal export contain GGAG motifs that are bound by hnRNP A2B1. We found this motif in some exosomal miRNAs but clearly not all, indicating that this motif is not a universal targeting signal.

D) ESCRTs vs. Sphingomyelinase. As shown in the Results and discussed in the Discussion (with references), we tested whether different biogenesis pathways might explain miRNA export selectivity. We indeed observed changes in miRNA signals in cells when treated with a sphingomyelinase inhibitor suggesting that there may be a distinct pathway for export. This only affected miRNA trafficking in mutant KRAS cells but not wild-type KRAS cells. While the data are intriguing, we were careful to qualify our results with the caveat that sorting in this situation might be cell-type or context-specific.

We agree that we have not solved the overall problem to understand or mechanistically explain miRNA export but we believe that our paper makes a significant contribution to the field by using our well controlled model system to test whether earlier proposed models hold up. Even though the data are negative, they provide a valuable tool for the field. We also include discussion of possible mechanisms that could explain selective export (see Discussion).

*2) Correlation analysis plays a central role in testing the authors' hypothesis. Given the low correlation of the independent biological replicates (deep sequencing replicates typically correlate with R > 0.90 in one of our labs), the authors should apply an appropriate statistical test to determine that R-values of 0.92-0.96 between cells are unlikely to differ from R-values of 0.67-0.89 comparing exosomes to the exporting cells*?

We thank the reviewers for this important comment. Variation between exosomes was obviously much larger than those between cells. In DESeq, the “per-condition” method is designed to handle this situation by calculating an empirical dispersion value for each condition separately. The “per-condition” method was the default option in an earlier version of DESeq. We were not aware of the change of the default to the “pooled” method in a more recent version, which was used to generate the results presented in the previous version of the manuscript. In the revision, we re-did differential analysis using the “per-condition” method and updated all related results accordingly (Tables 1 and 2, Figure 2—figure supplement 1, Figure 2—figure supplement 2, Figure 2—figure supplement 3 and Figure 2—figure supplement 4, Figure 3, and motif analysis, Figure 4—figure supplement 3). The new results do not alter the conclusions of the manuscript.

*If the bottom quartile of miRNAs by abundance (i.e., the ones least well measured by convention, rather than digital, sequencing methods) are excluded, do the Pearson correlation values change*?

Figure 2—figure supplement 2 shows new correlation values after removing the bottom quartile of miRNAs by abundance. These values are very similar to those calculated based on all miRNAs.

*Can all the biological replicates be used to make the comparisons, not simply pairwise combinations of individual data sets*?

Data from all biological replicates were used together in the differential expression analysis. The pair-wise analysis is just an exploratory analysis to gain a high-level overview of the pair-wise correlations of samples within or between different experimental groups.

*3) Is the degree of reporter repression small because the abundance of exosome-delivered miRNAs is low? The miRNA literature overwhelmingly supports the view that low abundance miRNAs have no biological effects, because the cellular concentration of miRNA-binding sites is much, much greater than the concentration of miRNA. That is, the stoichiometric mechanism of miRNA-mediated repression in mammals requires that miRNAs be highly abundant. When DKs-8 cells obtain a miRNA, such as* miR-222*, from exosomes, does that new miRNA rank in the top 25% or 50% of miRNAs by abundance? If not, it is difficult to imagine how it could be functional, given the aggregate intracellular concentration of seed-matched target sites. The authors need to report an estimate of how many molecules of a given miRNA sequence are present per exosome and how many are delivered to an individual recipient cell*.

The reviewers are indeed correct that the degree of reporter repression is small because the abundance of exosomal-delivered miRNA is low. This is not unexpected and is central to controversies in the field as to the stoichiometry and function of secreted RNA. Indeed, we think this is the key question moving forward because experimental proof of transfer can be difficult. Many studies successfully showing RNA transfer utilized experiments where nonphysiological concentrations of purified exosomes were added to recipient cells (see manuscript references). Evidence of transfer has also been demonstrated between immune cells that can remain opposed to one another for hours facilitating exRNA transfer but making it very difficult to precisely quantify the level of transfer (Mittelbrunn et al., Nature Communications 2:282, Ekstrom et al. JEV 1:18389, Montecalvo et al. Blood 119: 756). In our experiments, we chose to use Transwell co-culture experiments to resemble a more physiological system and also to test functional miRNA transfer with reporter constructs. We observed a ∼60-65% decrease with perfect sites and a ∼40% decrease with a wild-type mTOR 3’ UTR. Analysis of *miR-100* levels in recipient cells showed an approximate 34 % increase in *miR-100* levels compared to cells cultured in the absence of donor cells (Figure 6). The increase in *miR-100* levels is supported by precise copy number calculations that show that there are 329 molecules of *miR-100* per ng of total input RNA in cells grown in the absence of donor cells and those numbers increase to 443 molecules of *miR-100* in the presence of donor cells (Figure 6—figure supplement 2).

The statement that “overwhelming evidence supports the view that low abundance miRNAs have no biological effects” might be true on a global genomic scale but becomes a bit too generalized when applied to specific cells or developmental time points. There are two issues. First, one has to account for the concentration of a specific miRNA and second, the concentration of all target mRNAs in specific cells and/or at specific developmental time points. The fact that there have been about 11 different published target prediction algorithms speaks to the fact that we do not yet know precisely how to identify mRNA targets, nor do we know the exact set of rules that govern pairing and repression. Seed pairing is clearly important (Lewis et al. Cell 115: 787; Brennecke et al. PLos Biol 3: e85; Grimson, Mol Cell 27: 91; Lewis et al. Cell 120: 15; Krek et al. Nat Gen 37: 495) but there are lots of examples where imperfect seed sequences are robustly silenced (Li et al. NAR 36: 4277; Didiano and Hobert, NSMB 13: 849). Identifying bona fide targets requires experimental validation; in silico prediction is just a starting point with many false positive and negatives. Thus, knowing the exact concentration of target mRNAs and extending that to determine whether a miRNA is low abundance or not is not trivial. An historical example is *lsy6* in *C. elegans* which cannot be detected in sequencing approaches because it is so low abundance in worms, yet it controls the formation of left/right asymmetry between two neurons and regulates *cog-1* through a non-canonical binding sequence (Johnston and Hobert, Nature 426: 845). The effects of *lsy6* manifest themselves as part of a downstream network that is put in play by initial miRNA regulation, nicely illustrating the point that time and place matters. Our work on the role of miRNAs during vertebrate development has also identified numerous miRNAs that are not highly abundant but whose disruption leads to specific developmental defects (Flynt et al. Nat Gen 39: 259; Li et al. Dev 138: 1817; Thatcher et al. PNAS 105: 18384). Interestingly, we often find that non-canonical targets are the key targets even though we always begin our search using seed-based algorithms. Beyond our own work, perhaps the best example arguing against abundance and functional activity is *miR-33* which is expressed at approximately 30-40 copies per liver cell compared to *miR-122* which has about 3 million copies per liver cell. Despite the numbers, modest antagonism of *miR-33* down to about 15-20 copies per cell is enough to disrupt cholesterol metabolism and rescue atherosclerosis in mice (Najafi-Shoushtari et al. Science 328: 1566, Rayner et al. Science 328: 1570, Rayner et al. JCI 121: 2921, Rayner Nature 478: 404). Another good example is *miR-802* which is not even in the top 100 miRNAs as far as abundance in liver, yet it controls glucose homeostasis and type II diabetes (Kornfeld et al. Nature 494: 111). The Mendel lab had a nice paper using overexpression of *miR-26* in transgenic mice to try to resolve why *miR-26* can act as both a tumor suppressor and an oncogene in intestinal tumors. After overexpression, global analysis (GSEA) of likely mRNA targets identified numerous targets that were repressed by greater than 1.5. However, a known target of *miR-26* (EZH2) did not show up in their analysis yet they, and others, showed that *miR-26* regulates EZH2 (Zeitels et al. Genes and Dev 28: 2585). Lastly, *miR-206* was identified as one of several low abundance miRNAs that play key roles in colon cancer (Parasramka et al. Clin Epigenetics 4:16).

Nevertheless, we agree completely that stoichiometry is a key issue. The Tewari lab has published quantitative analysis of the amounts of miRNA per exosome and the numbers are startling low – 0.008 molecules of miRNA per exosome (Chevillet et al. PNAS 111: 14888). This illustrates the importance of the issue raised by the reviewers, something we completely agree with. Nevertheless, we observe silencing by *miR-100* and *miR-222* so despite stoichiometry issues aside, miRNAs are being functionally transferred.

*4) Why were three perfect sites used? Were controls performed validating the reporter using anti-miRs and miRNA mimics*?

Perfect sites were used in half of our reporter constructs to optimize detection of silencing. By creating perfect sites, we are inducing an RNAi-based Ago2 cleavage of mRNA targets which helps detect silencing through a catalytic mechanism. This provided the necessary proof of principle to try targeting an endogenous gene which was observed using the mTOR 3’ UTR. The binding sites in the mTOR 3’ UTR are typical miRNA binding sites with imperfect pairing so that repression is via miRNA-based mechanisms. Even with imperfect pairing, we were still able to observe silencing albeit less than when perfect sites were used. Further, mutation of the sites derepressed silencing. Three perfect sites were used (as opposed to one perfect site) because the endogenous mTOR 3’UTR also contains three *miR-100* sites. Having three perfect sites rather than one would actually underestimate the strength of Luc repression due to *miR-100* transfer.

The reviewers are entirely correct that we need to run extensive controls to ensure that the silencing we observe in our Transwell culture experiments is via *miR-100*. As requested, we performed antagomir experiments to decrease the expression of *miR-100* in donor cells (new Figure 6). We also included new data analyzing the absolute levels of *miR-100* in recipient cells grown in the presence or absence of donor cells (new Figure 6, Figure 6—figure supplement 2).

*5) In the ceramide experiments, the authors interpret the change in exosomal and cellular abundance for* miR-100 *and* miR-320 *as evidence that a subset of miRNA sorting is altered by ceramide while a separate, ceramide-independent pathway delivers other miRNAs to exosomes. The data are interesting, but don't seem to contribute to our understanding of the mechanism of putative sorting of miRNAs into exosomes. Perhaps* miR-10b *is simply less abundant than* miR-100 *or* miR-320*, making it harder to reliably detect changes in its abundance*?

The sphingomyelinase inhibition experiments begin to address mechanisms underlying the biogenesis of miRNA export into exosomes. In contrast to the statement by the reviewers, *miR-10b* is actually more abundant than *miR-100* or *miR-320* so levels do not appear to determine whether a miRNA is sphingomyelinase dependent or not. Overall, our findings are less about abundance and more about biogenesis and export of miRNA from cells. Controversy remains as to the varying roles of ESCRTs versus ceramide so it was important for us to demonstrate what we observe with our cells. As summarized in the Discussion, it seems that cell context is important and unifying conclusions are not yet possible for ceramide dependence.

*6A) High-Throughput Sequencing Data. How were the data normalized? How was the normalization procedure validated? Best practice is to select the normalization method that produces the greatest congruence among otherwise identical biologically independent replicates*.

The data were normalized using the DESeq package, in which the effective library size (i.e. size factor) for each sample is estimated using the function “estimateSizeFactors”. Dillies et al. (2013) evaluated several normalization methods based on real and simulated data sets (Dillies et al. Briefings in Bioinformatics 14.6 (2013): 671-683). Similar performance was observed for the DESeq normalization method and the TMM method, and both of them outperformed other methods.

*6B) Extending miRNA sequences {plus minus} 2 nt* “*to accommodate inaccurate processing of precursor miRNAs*” *would be a great idea if miRBase were always right; but miRBase is often wrong. It would be better to use the sequence of the most abundant isoform of the miRNA as the* “*accurately*” *processed form and to pool reads for all isoforms with the same 5′ end (i.e., the same seed sequence)*.

For each miRNA with a read count greater than 100, we compared the position of the most abundant isoform to the annotated position in miRBase. As shown in Table 3, consistency was found for around 75% of the miRNAs in all samples. Moreover, we compared miRNA counts based on miRBase annotations and positions of the most abundant reads, both with the +/- 2 strategy. As shown in Table 4, about 80% of the miRNAs had exactly the same counts and only about 5% of the miRNAs showed a difference of more than 10%. Based on these results, we decided to keep the miRBase-based counting results.

Author response table 1.Comparison of miRNA positions based on the most abundant reads and annotations from miRBase.**DOI:**
http://dx.doi.org/10.7554/eLife.07197.028SampleSameDifferentSame percentageDKO1.cell.12799674.4%DKO1.cell.22748676.1%DKO1.cell.32559073.9%DKO1.exo.11295271.3%DKO1.exo.21344375.7%DKO1.exo.3842875.0%DKS8.cell.133012572.5%DKS8.cell.229310873.1%DKS8.cell.32729075.1%DKS8.exo.1602174.1%DKS8.exo.2481675.0%DKS8.exo.3762476.0%DLD1.cell.12779275.1%DLD1.cell.230811572.8%DLD1.cell.32909774.9%DLD1.exo.1621778.5%DLD1.exo.2531676.8%DLD1.exo.3702176.9%

Author response table 2.Comparison of miRNA counts based on miRBase annotations and positions of the most abundant reads.**DOI:**
http://dx.doi.org/10.7554/eLife.07197.029SampleSameDifference more than 10%DKO1.cell.177.00%6.90%DKO1.cell.278.80%7.20%DKO1.cell.379.10%6.70%DKO1.exo.178.90%6.50%DKO1.exo.280.00%5.80%DKO1.exo.386.80%2.30%DKS8.cell.177.30%7.80%DKS8.cell.279.70%6.70%DKS8.cell.379.70%6.30%DKS8.exo.182.50%6.20%DKS8.exo.285.10%4.50%DKS8.exo.379.80%4.00%DLD1.cell.179.70%6.00%DLD1.cell.277.10%7.60%DLD1.cell.378.30%6.10%DLD1.exo.179.80%3.60%DLD1.exo.287.70%2.70%DLD1.exo.385.10%2.10%

*6C) Whenever read data is presented, species data should be presented in parallel. For example, the data in*
Figure 1
*would have a very different meaning if most of the* “*repeat*” *sequences were from just a few species, rather than a diverse set of RNAs*.

A count table for miRNAs is included in . Portions of reads from different repeat families are shown in Figure 1. A count table of the repeat families is also included in [Supplementary-material SD2-data].